

# Nadir ozone profile retrieval from SCIAMACHY and its application to the Antarctic ozone hole in the period 2003-2011

Sweta Shah[1], Olaf Tuinder[1], Jacob van Peet[1], Adrianus de Laat[1], and Piet Stammes[1]

[1]Royal Netherlands Meteorological Institute (KNMI), De Bilt, the Netherlands

*Correspondence to:* Sweta Shah (shah@knmi.nl) and Piet Stammes (stammes@knmi.nl)

**Abstract.** The depletion of the Antarctic ozone layer and its changing vertical distribution has been monitored closely by satellites in the past decades ever since the Antarctic "ozone hole" was discovered in the 1980's. Ozone profile retrieval from nadir-viewing satellites operating in the ultraviolet-visible range requires accurate calibration of level-1 (L1) radiance data. Here we study the effects of calibration on the derived level-2 (L2) ozone profiles and apply the retrieval to the Antarctic ozone

hole region.

We retrieve nadir ozone profiles from the SCIAMACHY instrument that flew on-board Envisat using the Ozone ProfilE Retrieval Algorithm) (OPERA) developed at KNMI with a focus on the stratospheric ozone. We study and assess the quality of these profiles and compare retrieved (L2) products from L1 SCIAMACHY versions 7 and 8 indicated as respectively (v7, v8) data from the years 2003-2011 without further radiometric correction. From validation of the profiles against ozone sonde

measurements, we find that the v8 performs better due to correction for the scan-angle dependency of the instrument's optical degradation.

The instrument spectral response function can still be improved for the L1 v8 data with a shift and squeeze. We find that the contribution from this improvement is a few percent residue reduction compared to reference solar irradiance spectra. Validation for the years 2003 and 2009 with ozone sondes shows deviations of SCIAMACHY ozone profiles of $0.8\% - 15\%$ in

the stratosphere and $2.5\% - 100\%$ in the troposphere, depending on the latitude and the L1 version used. Using L1 v8 for the years 2003-2011 leads to deviations of $\sim 1\% - 11\%$ in stratospheric ozone and $\sim 1\% - 45\%$ in tropospheric ozone. Application of SCIAMACHY v8 data to the Antarctic ozone hole shows that most ozone is depleted in the latitude range from $70°$S to $90°$S. The minimum integrated ozone column consistently occurs around 15 September for the years 2003-2011. Furthermore from the ozone profiles for all these years we observe that the value of the ozone column per layer reduces to almost zero at a

pressure of $100\,\mathrm{hPa}$ in the latitude range of $70°$S to $90°$S, as was found from other observations.

## 1 Introduction

Ozone ($O_3$) is one of the most important trace gases in our atmosphere. Stratospheric $O_3$ absorbs the dangerous solar ultraviolet (UV) radiation making it an important protector of life. A small amount of $O_3$ is found in the troposphere originating from air pollution and photochemistry - this ozone is considered a health-risk.





Daily ozone monitoring using satellites dates back to the late 1970s with instruments like the Total Ozone Monitoring Spectrometer (TOMS, 1979) and Solar Backscatter Ultra Violet (SBUV) instruments, and since the mid-1990s also by the full UV/VIS spectrum covering satellite instruments like the Global Ozone Monitoring Experiment (GOME, GOME-2) (e.g. Burrows et al., 1999; Munro et al., 2016), Scanning Imaging Absorption spectroMeter for Atmospheric ChartograpHY (SCIAMACHY)

(Bovensmann et al., 1999), and Ozone Monitoring Instrument (OMI) (Levelt et al., 2006), to name a few. These successions of instruments allow us to compare long term global ozone layer behaviour and cross-check the quality of the measured data. A long-term monitoring of the ozone trend layer is primarily driven by global measurements of total ozone time series of such satellite data.

The vertical profile of ozone has traditionally been measured by in-situ electrochemical instruments attached to balloons

(ozone sondes). Although ozone sondes provide the most accurate method for ozone measurement, they are limited in the heights they can reach (< 35 km), and their geographical coverage is limited to approximately 300 stations worldwide that provide weekly ozone sonde profiles, and very few stations with a higher than weekly measurement frequency. Satellite measurements provide an alternative means for obtaining globally vertical ozone profiles. In general limb and occultation mode satellite instruments can well resolve the vertical distribution in stratospheric ozone. However, they are limited in their hor-

izontal resolution, and they have no sensitivity to ozone in the middle and lower troposphere. An alternative approach is to use satellite measurements in nadir mode by high-resolution spectrometers in the thermal IR, like IASI and TES, and in the UV/VIS, like GOME, GOME-2 (e.g. Cai et al., 2012; van Peet et al., 2014; Keppens et al., 2015; Miles et al., 2015). OMI (e.g. Liu et al., 2010; Kroon et al., 2011), and SCIAMACHY.

The observation principle of nadir ozone profile retrieval in the UV/VIS is based on the strong spectral variation of the

ozone absorption cross-section in the UV-visible wavelength range, combined with Rayleigh scattering. The key here is that the short UV wavelengths (265-300 nm) are back-scattered from the upper part of the atmosphere whereas the longer UV wavelengths (300-330 nm) are mostly back-scattered from the lower part of the atmosphere. This transition in the ozone cross-section between 265-330 nm is useful in retrieving its vertical profile. Nadir UV and visible spectra provide better horizontal resolution in ozone although their observations can only be carried out in daytime. In the thermal infra-red measurements can

be done during both night and day.

SCIAMACHY had both limb and nadir mode capability. There have been several studies of ozone profiles using SCIAMACHY limb data (e.g. Brinksma et al., 2006; Mieruch et al., 2012; Hubert et al., 2017). Brinksma et al. (2006) found biases in ozone profile of stratosphere < 10%. Also in their analysis of limb scatter ozone profiles from 2002-2008, Mieruch et al. (2012) found the stratospheric ozone have a bias of ∼ 10% against correlative data sets and this bias increased up to 100% in the troposphere

for the tropics. Similarly Hubert et al. (2017) in their more recent study of limb profiles find that the SCIAMACHY ozone biases are about ∼ 10% or more in the stratosphere with short-term variabilities of ∼ 10%. There has been very little published work on ozone profile retrieval from SCIAMACHY nadir mode, probably due to calibration issues.

In this study, we focus on nadir ozone profile retrieval from SCIAMACHY and the impact of L1 calibration improvements. We evaluate the three most recent versions of the SCIAMACHY L1 product dataset (described in Sect. 2) on the basis of





**Table 1.** SCIAMACHY Level 1 (L1) data characteristics

| Wavelength range [nm] | Cluster number/Channel | Integration time [s] | spectral resolution [nm] |
|:---:|:---:|:---:|:---:|
| 265-282 | 3/1 | [0.125-1] | 0.22 |
| 282-304 | 4/1 | [0.125-1] | 0.22 |
| 304-314 | 5/1 | [0.125-1] | 0.22 |
| 314-321 | 10/2 | [0.125-1] | 0.24 |
| 321-330 | 9/2 | [0.125-1] | 0.24 |

Integration time ranges from 0.125 s to 1 s depending on the which pixel it is.

retrieved nadir ozone profiles from the SCIAMACHY UV reflectance spectra. The result of this paper shows the improved quality of the latest L1 dataset version.

The results presented here highlight the need for further corrections of the L1 data. However, a detailed study of radiometric bias corrections in L1 data is beyond the scope of this paper. The focus of this study is to analyse stratospheric ozone and this

study is done for almost the entire mission length of 2003-2011 where validation is done globally for the latest L1 version available (v8). We will focus exclusively on the study of ozone in the stratospheric region (100–10 hPa), but we will briefly comment on the accuracies we get for tropospheric region (1000 – 100 hPa). The OPERA retrieval algorithm is briefly reviewed in Sect. 2.2. In Sect. 3, the analysis of the slit function of the instrument is presented. Results on the ozone profiles, and the comparison between the level-1 datasets are shown in Sect. 4. This is followed by comparison to sondes in Sect. 5 for the most

recent dataset from years 2003-2011 spanning almost the entire mission. We apply the SCIAMACHY dataset in analysing the Antarctic ozone hole in Sect. 6. We discuss the possible effects of L1 radiometric bias corrections and applying slit function corrections in Sect. 7, and finally conclude in Sect. 8.

## 2    Instrument, data and methods

SCanning Imaging Absorption spectroMeter for Atmospheric ChartograpHY (SCIAMACHY) is space-borne spectrometer on

board ESA's Environmental Satellite Envisat  (Burrows et al., 1995; Bovensmann et al., 1999) with both horizontal (limb) and vertical (nadir) mode viewing design covering the wavelength range from 212 nm (UV) to 2386 nm (infrared, IR) spread over 8 channels. Launched in March 2002, its mission lifetime spanned until 2012 and we have level-1 data  (Lichtenberg et al., 2006) from the spectrometer from August 2002 until April 2012. In this paper we concern ourselves with the nadir data and in retrieving the vertical distribution of ozone using 265-330 nm UV-VIS continuous spectral data. Each nadir state is an

area on the Earth's surface defined by the scan speed of the nadir mirror across track direction and the spacecraft speed in the along track direction, the field of view (FoV) and the operation of the instrument. This gives typical ground pixel sizes (or equivalently nadir states) of 240 km × 30 km for an integration time (IT) of 1.0 s and 60 km × 30 km for an IT of 0.25 s (Gottwald and Bovensmann, 2011). Alternatively the nadir viewing corresponds to an instantaneous FoV (IFoV) of 0.045°





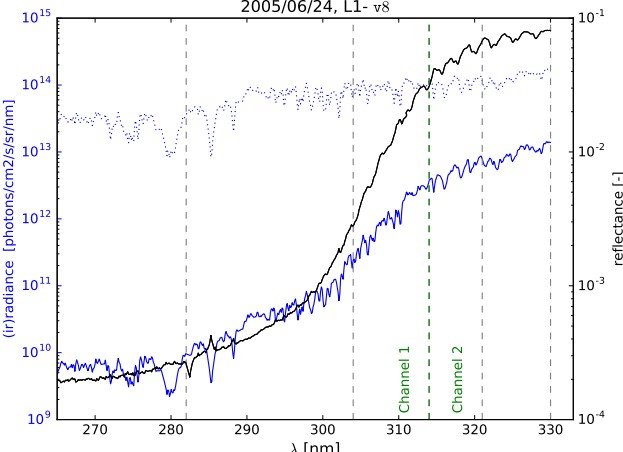

**Figure 1.** An example of the spectra of Earth radiance (solid blue line) and solar irradiance (dotted blue line) with the corresponding radiometric scale [$\mathrm{photons/cm2/s/sr/nm}$] indicated on the left side of the figure, and reflectance (solid black line) with the corresponding scale [unit-less] indicated on the right side of the figure. The vertical dashed line in green separates the two channels and vertical grey lines separates the clusters as listed in Table 1.

(across track) $\times$ 1.8° (along track). The IT also varies for clusters, the measurements should be combined before it is fed to the optimal estimation. The variation in IT between clusters can give rise to a spectral jump (discontinuity) where the last value of the spectral value in the preceeding cluster does not match the first value of the same in the following cluster. The wavelengths at which this occurs for Channels 1 and 2 were identified and blocked from our analysis (see Fig. 11 in Sect. 7).

5    The above specified wavelength range straddles over Channels 1 and 2 of SCIAMACHY with an overlap between the two channels. We use wavelengths 265–314 nm from Channel 1 and 314–330 nm from Channel 2. The extracted data from L1 are broken into states, which are groups of ground pixels. The spectrum of each ground pixel is divided into spectral clusters which are groups of wavelengths having their own integration time. These are then organized into clusters of data instead of channels. The mapping of clusters to wavelengths, the resolution, and IT are listed in Table 1.

10    The most important input for retrieving ozone profile are the Earth's reflectance spectra (from the L1 product). An example of these spectra are shown in Fig. 1. The observed reflectance spectrum is defined as:

$$R_{\mathrm{meas}}(\lambda) = \frac{\pi I(\lambda)}{\mu_0 E(\lambda)}, \tag{1}$$

where $I$, $\mu_0$, and $E$ are the radiance scattered and reflected by the Earth's atmosphere, the cosine of the solar zenith angle ($\theta_0$), and the incident solar irradiance at the top of the atmosphere perpendicular to the solar beam, respectively.



## 2.1 Versions of level-1 (L1) data

We make use of three different versions of L1 products (described below) from the nadir spectral data and present their differences using L2 products (ozone retrieval). Specifically the calibrated L1c data are reproduced using ESA's `SciaL1C` program of v3.2.6[1]. These versions we use in this paper are described below:

1. `v7`: L1 data products from SCIAMACHY are collocated spectra, radiometrically and spectrally calibrated radiances, including sun/moon occultation geometries. The version 7 (v7) specifically refers to $7.04 - W$ released in early 2012, where 'W' means processing stage flag and 7.04 is the software version number used for converting raw level-1 to L1 product (SCIAMACHY L1 processor, IPP $v7.04 - W$) (van Soest et al., 2005).

    2. `v7`$_{\mathrm{mfac}}$: This data set is identical to the one described above, except here we use the degradation corrections that were
provided independently as auxiliary data files[2]. Thus, the data structure allows us to turn on and turn off to check the effects of the degradation corrections independent of other calibration corrections. Degradation correction is obtained by the so-called 'm-factors'. The m-factors are determined by the monitoring of the light path which is given by the ratio of the measured spectrum of a constant source (Sun) to that obtained for the same optical path at a given time. This gives therefore an indication of the part of the degradation of the optical path as the instrument ages. These m-factors are
simple multiplication factors to the solar spectra after the absolute radiometric calibration (Gottwald and Bovensmann, 2011).

    3. `v8`: The IPP v8.02 is the 2016 version of SCIAMACHY L1 product. The main difference between this version with the ones above is the implicit implementation of a standard degradation correction. The degradation in this version takes into account the scan angle dependence of the nadir viewing geometry of the instrument with the optical path. We use the
slit-function key data provided in v8 for the instrumental slit function retrieval (see Sect. 3). Specifically the radiometric calibration uses a scan mirror model which takes into account the physical effect from the contamination layers in the mirror. The degradation using this model gives a scan angle dependence (Bramstedt, 2014).

## 2.2 OPERA retrieval algorithm

The Ozone Profile Retrieval Algorithm (OPERA) has been developed in KNMI (van Oss and Spurr, 2002; van der A et al.,
2002). It retrieves the vertical ozone profile using nadir satellite observations of back scattered UV sunlight from the atmosphere using UV and visible wavelengths. The algorithm makes use of the laws of radiative transfer in computing the top-of-atmosphere radiances given a number of atmospheric scattering and absorption parameters. The ozone absorption cross-section decreases from $265\,\mathrm{nm}$ to $330\,\mathrm{nm}$ which allows us to retrieve the amount of ozone as a function of atmospheric height. The retrieval method is based on a forward model with a maximum posteriori approach following Rodgers (2000). This amounts

---

[1]https://earth.esa.int/documents/700255/708683/RMF_0140_SCI

_NL__1P_v1.1_Dec2016.pdf/ba0b5e7b-f253-4e65-94b3-51fc23d1143d

[2]http://www.iup.uni-bremen.de/sciamachy/mfactors/



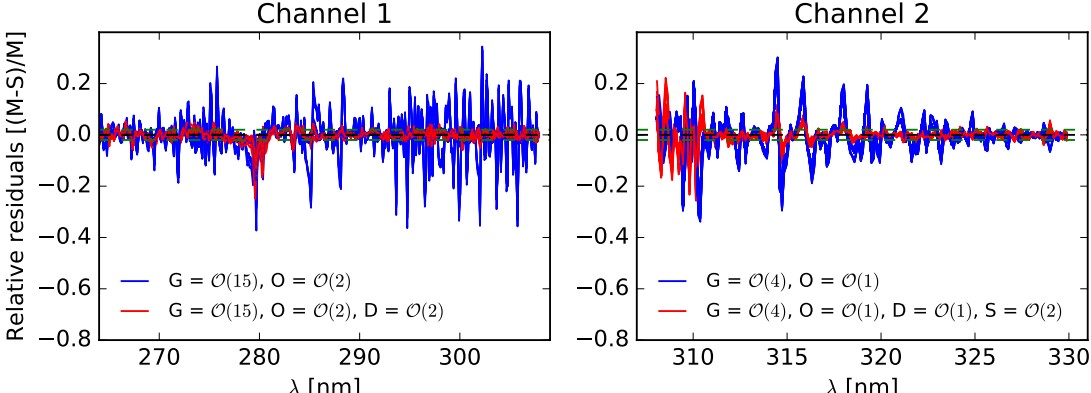

**Figure 2.** Relative residuals of daily SCIAMACHY L1 v8 solar spectra for Channels 1 and 2. G, O, D, S are Gain, Offset, Displacement/Shift and Squeeze respectively, where M is SCIAMACHY Measurement and S is Simulation (or modified model reference, see text). For visibility the residuals for $\sim 300$ spectra are shown spread throughout the mission length from August 2002 to April 2012.

to obtaining the state of the atmosphere by using the radiative transfer model and inversion technique iteratively till the model atmosphere matches the measurement. For a comprehensive algorithm overview and retrieval configuration, along with a description of the evaluation of the algorithm and the application to GOME-2 data, we refer to (Mijling et al., 2010; van Peet et al., 2014). The configuration chosen for all the retrievals are tabulated in Table 2. The retrieval grid or the vertical resolution of the nadir profile in OPERA can be chosen according to the Nyquist criterion. For SCIAMACHY data we find that setting the vertical grid to 32 layers or more gives the same value for the degrees-of-freedom (DFS). DFS (used in the Results section below) is a number related to the averaging kernels of the instrument or the sensitivity of the instrument with vertical height.

In practice, the measurement (reflectance spectrum) $R_{\mathrm{meas}}(\lambda)$ in Eq. 1 is prepared in the beginning of the OPERA algorithm, which is then passed to the Forward model. This model contains vertical atmospheric profiles, temperature, a priori ozone profile, geolocation, cloud data and surface characteristics. The forward model is used to compute simulated radiance at the top-of-atmosphere at wavelengths determined from the measured instrumental spectral data. This is further used to generate reflectances by using convolved simulated solar irradiance spectrum. The inversion step that follows is based on the Optimal Estimation method requiring measurement, simulation and measurement uncertainties in vector/matrix forms. An inversion using derivatives of reflectances with respect to the desired parameter to be solved is carried out until convergence is reached or until the maximum number of iterations is reached. For a comprehensive description of the flow of the algorithm we refer to the OPERA manual (Tuinder et al., 2014).





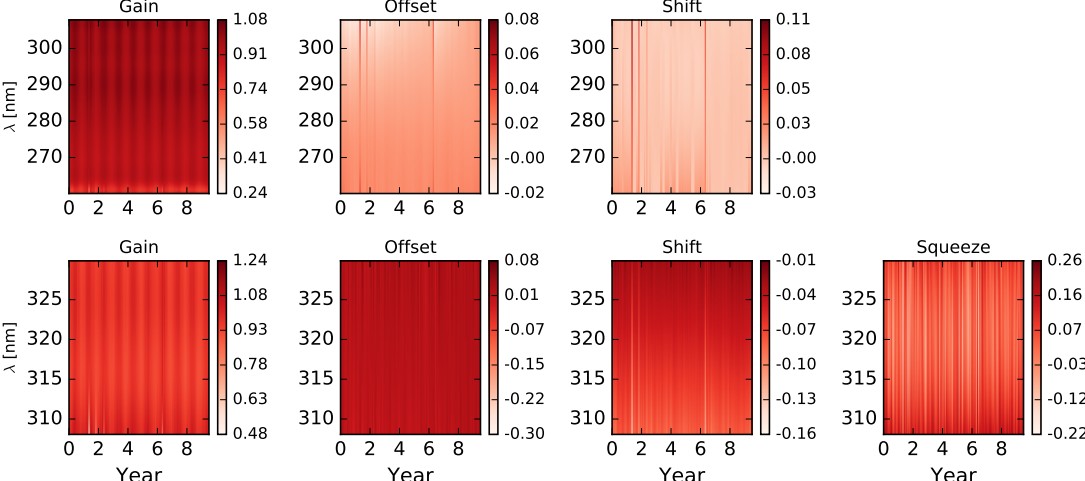

**Figure 3.** Temporal evolution of the slit function parameters shown for Channels 1 (top row) and 2 (bottom row). This is shown for the entire mission for 3466 days where Year 0 means 2002 and Year 8 means 2010.

## 3 Instrumental Slit Function calibration of the Solar spectral measurement

The accuracy of the retrieved geophysical product is primarily driven by the quality of the measured spectra, $R_{\mathrm{meas}}(\lambda)$ (Eq. 1), and its spectral and radiometric calibration. The most important SCIAMACHY specific calibration applied to the level-0 (raw data) is described in Slijkhuis et al. (2001). One of the spectral calibrations done often to assess the degradation of the

5   instrument in-flight is a fit of the instrument slit function (SF). It describes the behaviour of the projection of the incoming light onto the detector pixels. These were measured for SCIAMACHY on ground before launch (2002) and they are provided as L1 key data for the instrument that measures the solar spectra, $E(\lambda)$ (Eq. 1). The slit function of the instrument has different functional forms depending on which channel they belong to. For Channels 1-2 (relevant for our ozone retrieval) they are described by a single hyperbolic function:

$$\mathcal{S}(\lambda) = \frac{1}{a^2 + \lambda^2}. \tag{2}$$

The L1 key data provides the full width half maximum (FWHM) measured on ground which is related to the $a$ parameter in the equation above. This parameter can be solved in terms of the FWHM by using the fact that at the central point ($\lambda_0$): $S(\lambda_0) = 1/a^2$. Thus, $a = \frac{\mathrm{FWHM}}{2}^2$. At each given solar spectrum wavelength, $\lambda_i$, this functional shape is numerically computed between [-1,1] nm centred at $\lambda_i$. This shape can be manipulated with the SF parameters: additive constant (Offset, $O$),

15   multiplication factor (Gain, $G$), Displacement of the peak along wavelength (Shift, $D$) and expansion and contraction of the spectral peak (Squeeze, $S$). The high resolution solar spectrum (Simulation) from Dobber et al. (2008) is modified with these four parameters to best match the SCIAMACHY measured solar intensity (Measurement, $E(\lambda)$). First spectral parameters,





shift, $D$ and squeeze, $S$ are applied to Eq. 2:

$$\mathcal{S}'_{\mathrm{SD}}(\lambda) = \mathcal{S}(\lambda\,[1 - S(\lambda)] + D(\lambda)), \tag{3}$$

followed by radiometric parameters, gain $G$ and offset $O$:

$$\mathcal{S}'_{\mathrm{GOSD}}(\lambda) = [\mathcal{S}'_{\mathrm{SD}}(\lambda)G(\lambda)] + O(\lambda), \tag{4}$$

The unit of Shift, $D$ is nm and of Offset, $O$, Gain, $G$ and Squeeze, $S$ are numeric factors and all four parameters depend on wavelength. The parameters FWHM and $\lambda_0$ at each wavelength taken from the v8 L1 key data were identical for all the solar spectra throughout the mission. These values are given at certain wavelengths spread throughout Channels 1 and 2 and were interpolated for the wavelengths in between. An Optimal Estimation (OE) algorithm is used to solve for the best fit parameter values using the solar measurements after the launch. The retrieved values at the solar spectrum wavelengths are then also
applied to the Earth radiance spectra $I(\lambda)$ (Eq. 1) interpolated at the solar spectrum wavelengths.

Thus for each solar spectrum wavelength the parameter values are used in the retrieval algorithm. These best fit values are computed using the slit function model with the above mentioned four ways of manipulation. The manipulations are that each spectral peak of the model spectrum (reference) can be transformed with: Offset, Gain, Shift, and Squeeze. Each of these spectral manipulations can be modelled as a polynomial of order $n$ for each slit function of the instrument at the desired
channels. The OE was run using this model from which the best values of the Offset, Gain, Shift, Squeeze as a function of wavelength of Channels 1 and 2 are retrieved. The best fit value is checked by evaluating the relative difference between the Solar Irradiance Measurement and Simulation which is the relative residual. The relative residuals for the best fit $\{O, G, D, S\}$ are shown in Fig. 2. Each curve is a residual for one solar spectrum where the blue line is achieved by using only gain and offset in the model and the red line is achieved by including the shift and/or squeeze.
We find that in Channel 1, the wavelength range 265-308 nm is the optimal range for obtaining smallest residuals. However because the ozone profile retrieval algorithm requires data from 260 nm, we do the optimal estimation from 260-308 nm. The higher wavelength values around 314 nm in Channel 1 are known to have had calibration problems and therefore we use them only until 308nm where the slit function retrievals are still well behaved. The irradiances at 308-314 nm in Channel 1 failed to match the model spectrum. We ran the optimal estimation on all solar measurements of the SCIAMACHY mission for each
day which amounted to 3463 solar spectra. We convolved all spectra with {Offset, Gain, Shift, Squeeze} of polynomial orders of = $\{2, 15, 2, -\}$ and found that the cost function for the majority of cases reaches less than 1 (as expected) within 10 loops of retrieval in the Optimal Estimation routine. We find that using Shift in range 265-308 nm in Channel 1 reduces the residuals significantly looking at the left-panel in Fig. 2.

The residuals in Channel 2 are larger throughout. Dividing the relevant range of 308-330 nm into smaller ranges to get
smaller residuals did not reduce the residuals any further. For the optimal wavelength divisions, we found the least residuals given by the polynomial orders of the set of {Offset, Gain, Shift, Squeeze} = $\{1, 4, 1, 2\}$. In Channel 2, shown in the right panel, we find that using Shift, Squeeze in the range 308-330 nm reduces the relative residuals significantly. The relative errors of the observation in the wavelength range which we will use for ozone retrieval, 265-330 nm, are in the order of $10^{-5}$. The relative





residuals are in the order of a few percent. So we expect an error of a few percent to propagate into the ozone retrieval despite more accurate solar irradiances. There are anomalies in the residuals at around 279 nm, 280 nm and 285 nm. These are due to the strong MgI and MgII lines from the solar spectra and will not be used for ozone retrieval as indicated in Table 2.

Figure 3 shows the temporal dependence of all the slit function parameters for Channel 1 (top row) and Channel 2 (bottom row) as density plots where the value for each wavelength and each day of the year is shown. The ranges of the values are to the right to each figure. The seasonal dependence of solar radiation is observed as expected in the parameter Gain (first column of the figure) for both channels. The other parameters do not show any seasonal dependence over the mission time.

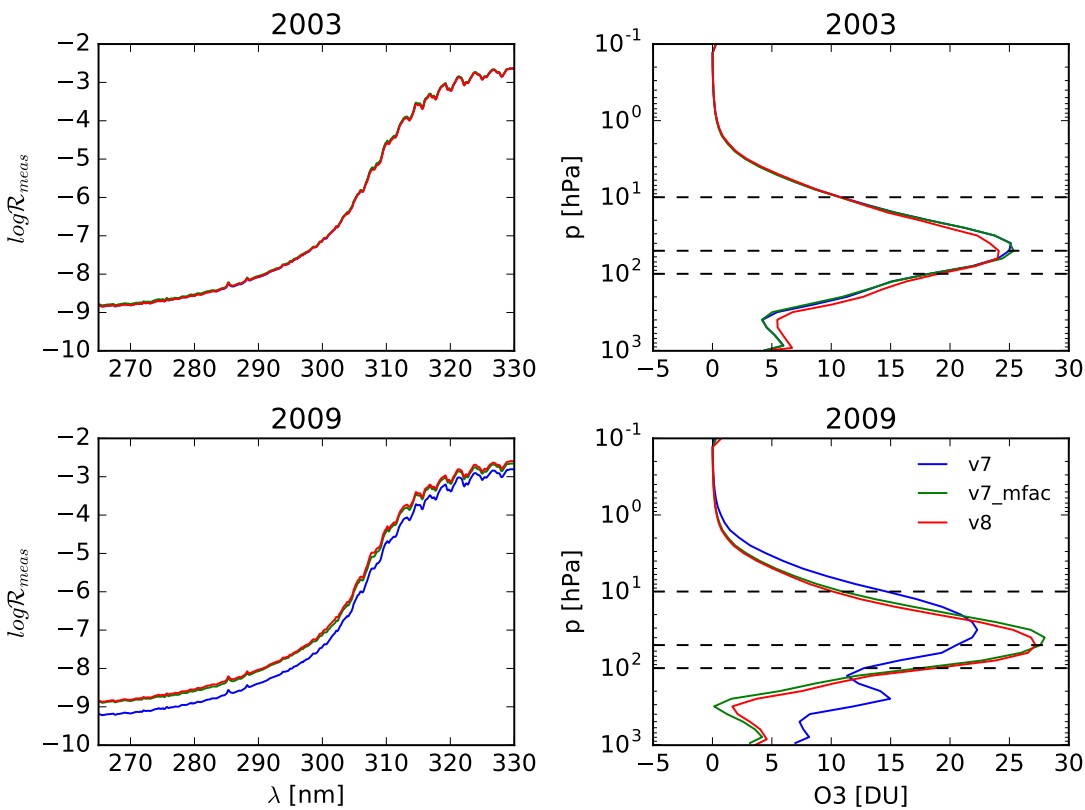

**Figure 4.** Top-left: Measured reflectance spectra for the year 2003. The spectra are an average over all the pixels in the range of the geolocations around the sonde stations within a narrow tropic latitude bands from $10°N$ to $10°S$. Top-right: Corresponding retrieved ozone profiles (in Dobson Units per layer) with different colours representing various versions of L1 SCIAMACHY data used (described in Section 2). The versions 7 with and without the degradation corrections almost overlap (blue and green lines) whereas version 8 with improved degradation correction has a visible difference in the ozone profile. Bottom rows are for the year 2009. Note the remarkable difference between v7 and later versions with degradation corrections. Here the later two versions almost overlap compared to the case where no degradation is taken into account showing the significant difference in L1 data between the three versions when degradation of the instrument is not taken into account. The number of pixels in each data set with corresponding uncertainties are listed in Table 3.



## 4   Results: ozone profiles for different L1 versions

We perform OPERA retrievals on SCIAMACHY nadir data for the entire mission length (2003-2011) on geolocations close
in space and time to ozone sondes. Here we use profiles in the vicinity of the sondes as a guidance for a general comparison
between the three datasets used. Thus the results presented in this section are for pixels (states) with global coverage for all

the months of the years 2003 to 2011. In Sect. 4.1 we show the comparison of the validation results for the three different
dataset versions for selected years. This is followed by validation results using v8 dataset for all the years, 2003-2011. The
main focus of this study is to analyse SCIAMACHY nadir ozone profiles in the stratosphere, using v7, $v7_{\mathrm{mfac}}$, and v8 L1 data
thus verifying their usefulness for ozone profile studies. With a nadir-viewing instrument it is very hard to retrieve an accurate
ozone profile in the troposphere which ranges in pressure height from $\sim 1000-100\,\mathrm{hPa}$. The lower-middle stratosphere ranges

from $\sim 100-10\,\mathrm{hPa}$ which is the main focus of this paper as motivated in Sect. 1 above. Thus in the subsections below we
will present a discussion of the quality of the SCIAMACHY nadir $O_3$ profiles in the lower-middle stratosphere and compare it
to the results from other existing studies.

### 4.1   Comparison of the ozone profiles using datasets v7, $v7_{\mathrm{mfac}}$, and v8

We make a direct comparison of nadir ozone profiles and their corresponding reflectance spectra (with converged retrievals)

between the different dataset versions. We show the comparisons for the years 2003 and 2009 to show how the differences in
the measurements (and therefore the profiles derived from them) vary from early to late in the SCIAMACHY mission. In Fig. 4
the results for 2003 are shown in the top panels and the results for 2009 are shown in the bottom panels. The left panels show the
measured reflectance spectra used by the OPERA retrieval algorithm in estimating the ozone profile shown in the right panels.
Each curve is a median of many pixels, 347 for each dataset version of year 2003 and $\sim 400$ for the year 2009. The result in the

figure is an average for all the pixels over a narrow latitude band in the tropics from $10°\mathrm{N}$ to $10°\mathrm{S}$. The different colours of lines
represent different datasets as labelled in the bottom right panel of the figure. The curves of ozone profiles in the top-right panel
for 2003 representing v7, $v7_{\mathrm{mfac}}$ (blue and green respectively) almost overlap each other showing minimal differences between
the two versions and therefore the degradation corrections (m-factors) are also minimal. However, the curve representing v8
in red deviates from the other two visibly, which is hard to see in the reflectance spectra in the left panel. These differences

are exacerbated for the year 2009 (later time of the mission) where the profile of v7 is significantly different from $v7_{\mathrm{mfac}}$ and
v8 in its shape and amount of ozone. The corresponding measured spectra in the bottom-left-panel confirm these differences.
We also observe visible differences between $v7_{\mathrm{mfac}}$ and v8 indicating the intrinsic differences in the implementation of the
degradation corrections between the two dataset versions. In the right panels of the figure are horizontal black-dashed lines
demarcating the lower-middle stratosphere $(100\text{-}10\,\mathrm{hPa})$ with another line at $50\,\mathrm{hPa}$. In dataset v7 there are large variations

in the troposphere $(1000\text{-}100\,\mathrm{hPa})$ and a significant reduction of the peak of the ozone value in the stratosphere, suggesting
the unreliability of this dataset for later years of the SCIAMACHY mission. The median errors and standard deviations (st.
dev.) along with number of pixels and convergence statistics for the retrievals in Fig. 4 are listed in Table 3. The maximum





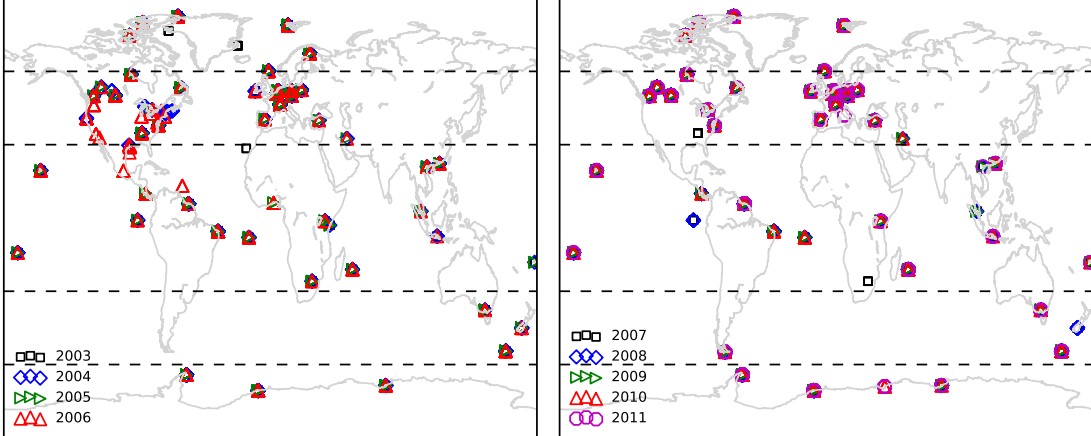

**Figure 5.** Collocated geolocations indicating the location of the ozone sondes used in the validation for the dataset of L1 v8. Left: collocated geolocations for years 2003-2006 as labelled. Right: the same for years 2007-2011 as labelled.

number of iterations, `n_iter` (see Table 2), is set to 10 and the median values of this quantity in the table conversions. This also suggests that further corrections to the L1 data are needed (See Sect. 7).

## 5 Validation: Comparison with ozone sondes

For validation, the retrieved ozone profiles are compared with balloon ozone sondes obtained from the World Ozone Ultraviolet
5  Radiation Data Centre (WOUDC, 2011). The sonde is used if it is located within the four corners of the SCIAMACHY pixel/state on the ground and has a measurement date and time within 6 hours of the sonde.

The resulting geolocations of the selected sondes from the WOUDC data set are plotted in Fig. 5 for all years. The validation algorithm implementing the collocation criteria is very similar to the one used by van Peet et al. (2014). For the methodology we refer to that paper. The number of stations for years 2003-2011 range from 38 to 66 with number of sondes ranging from
10  1 to 55 per station. In comparing the retrieved profile with that of the sonde, the sonde profile is convolved with the averaging kernel (AK) from the retrieval characterizing the sensitivity of SCIAMACHY at each layer. This gives a smoothed sonde profile which is more suitable to compare with the profile retrieved from the satellite instrument and influences the results in Figs. 6-7 which are discussed below. To get an impression of the shape and effect of the averaging kernels, we refer to Appendix A. There in Figure A1 we show an example of SCIAMACHY averaging kernel shapes for a subset of the retrieved layers. In
15  Figure A2 we show the validation results for the case where the sondes were convolved with the averaging kernel and the case where the sondes were not convolved.





## 5.1 Validation comparison between versions $v7$, $v7_{mfac}$, and $v8$

In Fig. 6, we show the validation for the years 2003 and 2009 and explicitly show the differences in the three datasets. The top row panels show the validation for the Southern Hemisphere (SH) with latitudes band $90°$S to $30°$S, middle row panels show the same for Tropics (Tr) with latitude band $30°$N to $30°$S and finally bottom row panels show validation for the Northern Hemisphere (NH) with latitude band $30°$N to $90°$N. Different data sets: $v7$, $v7_{mfac}$ and $v8$ are shown in dashed, dash-dotted and solid lines, respectively, as labelled. For reference the errors (25%-75% percentiles) in the relative differences are shown only for $v8$ for all the latitude bands and both years as shaded regions. Each curve is the median difference between retrieved ozone layer value and the averaging-kernel smoothed ozone sonde value, normalized by the ozone sonde profiles. The lower-middle stratospheric region is marked by yellow-dashed lines where better quality agreement of the nadir ozone data is expected with the sonde. An ideal agreement between sonde and satellite would give a difference of zero at all layer heights.

The relative percentage differences between the nadir profiles and the sondes are given in Table 4, where the top half of the table shows the results for the years 2003 and for datasets $v7$, $v7_{mfac}$ and $v8$ for each zone: SH, Tr and NH. The same is listed in the bottom half for the year 2009. From the table we see that the absolute values of the deviations (see third column, st. dev. [%]) in the stratosphere are systematically smaller than in the troposphere (see fifth column). These large spreads in st. dev. of satellite retrievals from sondes are also visible in Fig. 6. Any deviation above $\sim 15\%$ in the stratosphere and above $20\%$ in the troposphere are shown in bold for reference as these are the required accuracy levels in the ESA CCI programme (http://www.esa-ozone-cci.org/). This is probably not due to the relatively worsening sensitivity of the nadir instrument to the tropospheric retrievals (van Peet et. al., *private communication*). Rather it might be due to the quality of the SCIAMACHY data. Deviations in validation in the upper troposphere and lower stratosphere have been reported due to the ozone variability in a previous study of GOME-2 nadir data (Cai et al., 2012). However, in the stratosphere (within the yellow dashed lines), the median deviations for 2003 are smaller for $v8$ for Tr and NH compared to the older dataset versions, whereas for the SH the three different datasets give comparable deviations. The deviations for 2009 in $v8$ in the stratosphere are smaller for SH and NH than in the older datasets; the deviations are comparable in the Tr zone between all datasets. Comparison of the deviations between 2003 and 2009 show larger values for 2009 suggesting that the quality of L1 $v8$ data has degraded and is much worse than for earlier years (from comparison with 2003 $v8$ data).

## 5.2 Validation of $v8$ for years 2003-2011

In Fig. 7 we show validation results for the entire dataset of $v8$ from the years 2003-2011 organized according to latitude bands with top panels showing results for SH, middle panels for Tr and bottom panels for NH. The left column show results for (early) years 2003-2006 and right column for (later) years 2007-2011. The solid lines correspond to the difference between satellite and sonde (convolved with satellite AK) normalised by the sonde profile and the dashed lines correspond to the difference with the a priori profiles used in the retrieval (see Table 2). The number of collocated pixels for all the latitude bands for each year is listed in Table 5 along with median degrees-of-freedom (DFS, see Sect. 2.2), median number of iterations required to achieve convergence, the median solar azimuth angle and the median deviations in % for each zone in stratosphere and troposphere in





the last two columns. From Fig. 7 we note that validation for all latitude bands show smaller deviations from the centre (zero) line for earlier years (left column), and for later years (right column) the agreement with sondes become worse especially for tropics and NH (right middle and bottom panels). The median deviations in the sixth and seventh columns of Table 5 show often values higher than $20\%$ (in boldface) for troposphere whereas these values are less than $15\%$ for the stratosphere

deeming them within specifications according to ESA CCI requirements (see Sect. 4.3). It should be noted however that the large deviations in the tropospheric ozone $[1000\text{-}100]\,\mathrm{hPa}$ are systematically higher (than those for the for stratosphere) for all zones and the years even for v8, where such deviations are not observed for instance with GOME-2 nadir profiles (van Peet et al., *private communication*). This suggests that the quality of nadir SCIAMACHY L1 data is still poor and can be improved upon. It affects the lowest troposphere in the beginning of the mission and gets worse due to instrument degradation. This is

still uncorrected in the UV wavelength range. Also observe the higher deviations in the year 2003 in SH and Tr (left upper and middle panels) compared to the rest of the years. This unique behaviour of 2003 is discussed in Sect. 7.

The deviations in SCIAMACHY v8 validation results above for the stratosphere can be compared for instance with GOME-2 validation results in van Peet et al. (2014) where they have used 16-layer pressure grid layer for retrievals. Their validation in the troposphere also showed deviations ranging from a few percent to $\sim -30\%$ for the Northern Hemisphere, whereas we

find the deviations to range from $\sim -10\%$ to $-45\%$) for year 2008 (blue line in right panel Fig. 7. In the Southern Hemisphere however, we find these deviations for year 2008 (top right panel Fig. 7) to range from a few percent to $\sim 20\%$ which is more comparable to the range of a few percent to $\sim -15\%$ in van Peet et al. (2014).

## 6 SCIAMACHY results of the Antarctic Ozone Hole

In this section we show an application of the SCIAMACHY dataset using L1 v8 to infer the ozone in the Antarctic region.

We apply the OPERA retrieval algorithm to all ground pixels south of $45°$ for the years 2003-2011 only for the months of (beginning of) August to (end of) November. The retrievals are separated in three latitude bands: $[45°\mathrm{S}{:}55°\mathrm{S}]$, $[55°\mathrm{S}{:}70°\mathrm{S}]$ and $[70°\mathrm{S}{:}90°\mathrm{S}]$ shown in the top, middle and bottom rows of Fig. 8 respectively. The colour represents which year is labelled, showing early years (2003-2006) and later years (2007-2011) in the left and right columns of the figure, respectively. Each circle in the figure is the minimum value of the retrieved column for each day. The size of circle represents the number of

pixels averaged per day and the range (minimum, maximum) of the number of pixels for all years and all latitude bands are listed in Table 5. The median uncertainties in the retrieved columns and their st. dev. are also listed in Table 5. The variations in the minimum integrated ozone are lowest for the latitude band: $[45°\mathrm{S}{:}55°\mathrm{S}]$ for all years (see top row of the figure) instead of a V-shaped dip appears in the southernmost latitude bands (see middle and bottom panels in the figure). This is as expected as the ozone depletion is stronger in the Antarctic region. Also the time of the minimum of the daily minimum integrated ozone

columns occur between 15 September - 15 October which is also expected from other published results.

We compare these time series with the Multi Sensor Reanalysis of Ozone, version 2 (MSR v2) (van der A et al., 2015). The MSR uses data from TOMS, SBUV, GOME, SCIAMACHY, OMI, GOME-2 for latitudes south of $30°\mathrm{S}$ resulting in a multi-decadal ozone column. The reprocessed ozone columns from all satellites are assimilated and bias corrected by calibrating with





ozone columns obtained from Brewer and Dobson spectrophotometers in the WOUDC dataset. The SCIAMACHY L1 dataset included in the MSR v2 makes still use of v7 using the the total nadir ozone retrieval algorithm (TOSOMI) (Eskes et al., 2005; Valks and van Oss, 2003). The minimum ozone value available for any day from the MSR dataset is plotted for latitude band of 70°S:90°S in Fig. 9 and can be qualitatively compared to the bottom two panels in Fig. 8. The characteristic "V" shape in the

Fig. 9 is in line with the minimum occurring between 15 September to 15 October for the years 2003-2011. The lowest value in the SCIAMACHY ozone in Fig 8 occurs between 1 September and 15 October for latitude bands of 55°S:70°S and 70°S:90°S in the middle and bottom panels. The ozone columns reach a plateau in Fig. 8 from 1 November for latitude band 55°S:70°S where as this feature is not visible for the band 70°S:90°S which is more consistent with the Fig. 9. The overall level of the ozone column is at $\sim 150$ in September for 70°S:90°S for the years 2004-2011 which is also qualitatively consistent with the

MSR dataset.

It is useful to specifically compare the SCIAMACHY ozone total column in the stratosphere to the MSR v2 dataset for the year 2010, which was found to be an anomalous year in its behaviour of the ozone depletion (de Laat and van Weele, 2011). The Antarctic ozone hole had 40-60% less ozone destruction compared to the average of previous years (2005-2009). This is reflected in the right-panel of Fig. 9 where the minimum integrated ozone (in red) are above the rest of the years, showing

higher levels of ozone from 1 August to 1 November with variations in the range of 250-150 DU. Comparing this with the right panel of Fig. 8 we note that SCIAMACHY nadir profile data does not pick up this anomalous behaviour of 2010 (also in red circles), the values in average are much lower than those in Fig. 9 and at most comparable to the rest of the years in Fig. 8 with variations in the range of 170 -100 DU. This could suggest that SCIAMACHY nadir profiles using plain L1 v8 data on its own may not be accurate enough (owing to remaining biases in L1 data, other instrumental issues like the instrumental

coverage) to investigate inter-annual variability in Antarctic ozone depletion. Note that there is no SCIAMACHY data in early August south of latitude 70° due to the low Sun. Furthermore note that the MSR v2 does a time dependent bias correction before assimilation takes place. However, in the region of 55°S:70°S (mid-right panel of Fig. 8 we observe that the minimum ozone column for 2010 does exhibit the anomalous behaviour where it is higher compared to the rest of the years. This latitude region includes 70°S, which samples the vicinity of the outer edge of the ozone hole, this result does show that SCIAMACHY

nadir profiles could be used to complement the inter-annual variability studies. The discrepancy between the minimum of the total ozone column in regions 55°S:70°S and 70°S:90°S can be further investigated by carrying out a bias study of the L1 data (see Sect. 7).

In Fig. 10, the ozone profiles are shown for various latitude bands as in Fig. 8. The median of the profiles for each latitude band and for each year is plotted as labelled with same colour coding as in Fig. 8. In general the maximum of the ozone for all

years decreases with the southernmost latitude bands going from top to bottom of each column in Fig. 8. The peak value of the ozone (maximum) decreases from $\sim 30$ DU at latitude band of 45°S:55°S to $\sim 25$ DU at the latitude band of 70°S:90°S for all the years. Furthermore, the curve tends to be bi-modal at the southernmost latitude band with a minimum value of ozone (a value of zero DU) at a pressure of $100\,\mathrm{hPa}$. This is the expected height of ozone depletion. The number of pixels for each year and for each latitude band are listed in Table 6 along with the median uncertainties in the ozone profile along with its

spread. Columns 5-7 in Table 6 list the median number of iterations to reach conversion and the cost function that measures



the deviation of simulated and measured spectra at n-th iteration. What can be seen is the increasing number of iterations and corresponding increase in the cost function (thus worsening retrievals) for the southernmost latitude bands.

## 7 Discussion

In the previous section the ozone hole in the Antarctic was analysed using SCIAMACHY data for years 2003-2011 using geolocations south of $45°$. It is evident from Fig. 8 that the integrated ozone for the year 2003 (shown in black open circles) in the left panels are outliers. The retrieved columns deviate significantly in the latitude band $55°S:70°S$ (middle row) in the month of August and the deviations are significant for the same year in latitude band $70°S:90°S$ (bottom row) for all the months. Furthermore, the total ozone columns of the year 2004 are similar to those of 2003 compared to the other years in the bottom row of Fig. 8. This behaviour is reinstated further in Fig. 10, in the left - middle and bottom rows where the ozone profiles in the those bands for years 2003 and 2004 show a significant deviation from the rest of the years. In fact, the southernmost ozone profiles also show that the deviations are strong for the years 2004, 2005. From the assimilated and calibrated MSR v2 dataset (van der A et al., 2015) shown in Fig. 9 there are no significant deviations of ozone columns for 2003 (left panel). In a study by (Tilstra et al., 2012), it was shown from the retrieval of Absorbing Aerosol Index (AAI) using SCIAMACHY nadir spectra from 340- 380 nm that strong jumps in the neighbouring AAI were observed from day to day in the years 2003, 2004 and 2008. These are exactly correlated with the days where the instrument was heated up to get rid of the ice layer affecting the infrared wavelengths and their degradation. Probably this operation on SCIAMACHY instrument affects all the versions of L1 data. We suggest that these instrumental throughput changes are the cause of the deviating ozone profiles for the years 2003-2004.

In Sect. 3, it was shown that spectral corrections like shift and squeeze at the UV wavelengths can further improve the solar spectra by $\sim 4\%$ (see Fig. 2). However improvement from this is much less than the expected degradation and other remaining potential biases for the L1 data which are increasing with the mission time. A preliminary analysis of this is shown in Fig. 11, where the mean ratio of observed to simulated reflectance spectra ($R_{meas}/R_{sim}$) is plotted for the day of June 24 for years 2003-2011 in fading black to white colour. There is a strong deviation of this ratio at 265-290 nm and this becomes worse for later years, 2008-2011. A detailed study of this bias is beyond the scope of this paper. However, a L1 reflectance bias correction can significantly improve the quality of L1 data and will influence the validation and other results in this paper from their ozone retrievals. Furthermore applying these bias corrections would also make it meaningful to apply the spectral slit function corrections that can potentially improve the ozone retrievals further. Thus a detailed study of the effect of such bias corrections on the L1 data can be investigated against the quality of the ozone retrieval algorithm to better understand the quality of the SCIAMACHY nadir data.



## 8    Conclusions

We have performed nadir ozone profile retrieval using the OPERA algorithm for the latest complete SCIAMACHY dataset (v8)
for almost the entire mission length from 2003 to 2011. Differences between datasets with and without degradation corrections
(m-factors) were analysed in the wavelength range of 265-330 nm and show that the degradation correction including the scan-
angle dependence in the L1 v8 dataset gives the most smooth ozone profiles which is also reflected in their validation against
ozone sondes. Retrieving the instrument slit function in the UV range for this L1 data set also gives improvement of a few
percent in the solar data through the mission length. However, the measured reflectance spectra show that the degradation in
v8 is still significant because the ratio of the measured to simulated reflectance spectra (calculated from the radiative transfer
model used in OPERA) can range from $\sim 1.1 - 1.4$ (see Fig. 11, Sect. 7). Furthermore the comparison between different L1
versions shows that v7 gives significantly worse ozone profiles especially later in the mission (2009) compared to v7$_{\mathtt{mfac}}$ and
v8, where the profiles v7 show a double peak for the year 2009 and over-all reduced amount of ozone. Thus L1 v8 should be
used for the nadir ozone profile applications of SCIAMACHY data.

Using all L1 v8 data below $45°$S for years 2003-2011 we investigated the Antarctic ozone profile behaviour in the austral
spring season. The daily minimum of the total ozone column from August to end of November shows a characteristic "V"
shaped curve with the dip in mid-September to beginning of October at the southernmost latitude bands. This is consistent
with other satellite data sets (for example, (van der A et al., 2015)). The outliers are the years 2003-2004, which can also be
seen in the profiles for various latitude bands averaged over the whole year. The overall peak value of ozone reduces with
southernmost latitude bands and a prominent minimum in the ozone profile with vanishing ozone concentration appears at a
height of 100 hPa, as expected.

**Appendix A:  Averaging Kernels from OPERA retrievals using SCIAMACHY data**

The averaging kernel (AK) of a retrieval represents the measurement sensitivity with respect to the true state of the atmosphere.
The rows of the ozone profile averaging kernel matrix give the smoothing of the true profile as a function of the ozone retrieval
layers. For an ideal retrieval, the curve of each row will peak at the nominal layer height with a spread that gives the vertical
resolution of the retrieval.
In Fig. A1 we show an example of the AK for an individual OPERA ozone profile retrieval for a pixel on 2004/01/07. In
Fig. A2, we show the effect on the validation of including the satellite AK to the ozone sondes for the years 2003 (left panel)
and 2009 (right panel). The validation results are clearly less noisy and smoother for the case where the AK was applied to the
ozone sondes.

*Acknowledgements.*  The authors would like to acknowledge the SCIA-Visie project, funded by the Netherlands Space Office (NSO). The
authors would also like to thank Gijsbert Tilstra for support with SCIAMACHY data. The authors would like to thank all data contributors to



the World Ozone and Ultraviolet Radiation Data Centre (WOUDC) for submitting their data to a public database and Environment Canada for hosting the database.



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





**Table 2.** Parameter settings in the OPERA algorithm for the $O_3$ profile retrieval

| Physical parameter/physics | Description | Setting used in OPERA |
|---|---|---|
| Model | LIDORT-A | van Oss and Spurr (2002) |
| | number of streams: LIDORT-A | 6 |
| Raman scattering | on or off | off |
| $O_3$ absorption cross-sections | temperature parametrised database using five temperatures for the polynomial temperature parametrised expansion | Malicet et al. (1995, e.g.) |
| | Brion, Daumont and Malicet cross-section | |
| Climatology | O3 a priori profile has three different ozone climatologies | Bhartia and Wellemeyer (2002); Fortuin and H. Kelder (1998); McPeters et al. (2007) |
| Temperature | temperature profile: ECMWF | |
| Noise floor | systematic relative error of measured reflectance | 0.015 (for all datasets) |
| Retrieval method | Optimal estimation | Rodgers (2000); van der A et al. (2002) |
| Wavelength window | variable bands (can be set independent of instrument channels) | [265 - 330] nm |
| | blocked MgI, MgII lines | [284.5 - 286.5] nm, [278.0 - 280.5] nm |
| Maximum number of Iteration | convergence based on decrease of the relative cost function | 10 |
| Pressure grid | number of layers heights for retrieval | 32 |
| Fit parameters | ozone column per layer | |
| | Fit albedo | for cloud fraction < 0.20, fit for surface albedo else fit for cloud albedo |
| Cloud model | parameters: effective cloud fraction height, pressure profiles | FRESCO+ (Wang et al., 2008) |
| | surface albedo | Koelemeijer et al. (2003) |



**Table 3.** Median uncertainties in the level-2 product statistics

| Year | version | # Pixels | n_iter | DFS | $\langle \sigma^{R_{\mathrm{meas}}}/R_{\mathrm{meas}} \rangle$ | $\mathrm{std}(\langle \sigma^{R_{\mathrm{meas}}}/R_{\mathrm{meas}} \rangle)$ | $\langle \sigma_{O3}/O3 \rangle$ | $\mathrm{std}(\langle \sigma_{O3}/O3 \rangle)$ |
|---|---|---|---|---|---|---|---|---|
| 2003 | v7 | 347 | 5.0±2.3 | 5.5±0.6 | 0.015 | 0.060 | 0.072 | 0.074 |
| 2003 | v7$_{\mathtt{mfac}}$ | 347 | 5.0±2.3 | 5.5±0.6 | 0.015 | 0.059 | 0.075 | 0.076 |
| 2003 | v8 | 347 | 5.0±2.0 | 5.5±0.6 | 0.015 | 0.013 | 0.072 | 0.066 |
| 2009 | v7 | 392 | 6.0±1.9 | 4.9±0.4 | 0.052 | 0.239 | 0.056 | 0.044 |
| 2009 | v7$_{\mathtt{mfac}}$ | 408 | 7.0±2.2 | 4.9±0.5 | 0.051 | 0.239 | 0.076 | 0.130 |
| 2009 | v8 | 409 | 5.0±2.2 | 4.9±0.4 | 0.054 | 0.260 | 0.082 | 0.124 |

*Column 1*: L1 data year, *Column 2*: dataset version, *Column 3*: Number of states/pixels used in computing median quantities, *Column 4*: median n_iter for # pixels ± standard deviation, *Column 5*: median degrees-of-freedom (see Sect 2.2) ± standard deviation. *Column 6*: median relative uncertainty of converged measured reflectance spectrum in the wavelength band used, *Column 7*: standard deviation of *Column 7*, *Column 8*: relative uncertainty in retrieved ozone profile, *Column 9*: standard deviation of *Column 8*





**Table 4.** Validation statistics of various dataset versions

|  | Stratosphere [1000-100] hPa | st. dev. | Troposphere [100-10] hPa | st. dev. |
|---|---|---|---|---|
| Zone | Range | [%] | Range | [%] |
| **2003** | v7 |  |  |  |
| SH | [-6.2  2.1 ] | -1.0 | [**-38.4**  -6.2] | **-12.7** |
| Tr | [-1.6  **17.7**] | +8.3 | [**-71.9 13.5**] | **-51.6** |
| NH | [-2.5  5.2] | +2.9 | [**-39.8**  -2.5] | -11.2 |
|  | v7$_{\text{mfac}}$ |  |  |  |
| SH | [-6.5  1.3] | -1.4 | [ -8.9  -0.3] | -3.8 |
| Tr | [-1.3  **17.0**] | +8.9 | [**-72.2 12.6**] | -53.6 |
| NH | [-2.4  4.7] | +2.7 | [**-35.2**  -2.4] | -11.1 |
|  | v8 |  |  |  |
| SH | [-9.4  3.9] | -5.0 | [ 3.9  **43.1**] | **+12.4** |
| Tr | [-2.5  **13.2**] | -1.6 | [ 13.2  **51.0**] | **+40.4** |
| NH | [ 0.0  2.4] | +0.8 | [ -0.5  4.0 ] | +2.5 |
| **2009** | v7 |  |  |  |
| SH | [**-16.4 28.9**] | -4.7 | [ -1.7  **100.7**] | **28.1** |
| Tr | [**-33.5 16.8**] | **-13.3** | [ **-23.1 220.0**] | 179.1 |
| NH | [**-25.5 35.8**] | -0.9 | [ **-25.5  72.5**] | **26.1** |
|  | v7$_{\text{mfac}}$ |  |  |  |
| SH | [-8.8  **15.2**] | +2.2 | [ **-66.9**  -8.8] | **-29.8** |
| Tr | [ 3.6  **18.5**] | **+12.3** | [**-109.0 10.3**] | **-84.7** |
| NH | [-4.1  14.1] | +8.4 | [ **-99.4**  -4.1] | **-30.6** |
|  | v8 |  |  |  |
| SH | [-9.7  4.0] | +1.3 | [ **-19.2**  0.4] | -5.8 |
| Tr | [-4.7  **36.7**] | **+11.6** | [ **-63.0 36.7**] | -37.8 |
| NH | [-2.1  10.7] | +6.3 | [ **-71.8**  2.4] | **-22.9** |

*Column 1*:Year followed by latitude zone, *Column 2*: [Minimum Maximum] of $\left\langle \frac{a_{\text{opera}} - a_{\text{sonde}}}{a_{\text{sonde}}} \right\rangle$ [%] for stratosphere, *Column 3*: Median of $\left\langle \frac{a_{\text{opera}} - a_{\text{sonde}}}{a_{\text{sonde}}} \right\rangle$ [%] for stratosphere, *Column 4,5*: Same as in *Column 2,3* for troposphere




**Figure 6.** Comparison of validation for three different dataset versions as labelled in the bottom right panel of the figure: dashed line is v7, dash-dotted line is v7$_\text{mfac}$ and solid line is v8. The validation is shown for Northern Hemisphere (NH), Tropics (Tr), and the Southern Hemisphere (SH) from top to bottom rows. For reference the uncertainty in the difference between the retrieved profile and the convolved sonde at each layer is shown for the v8 dataset.





**Figure 7.** Validation of ozone profiles retrieved using OPERA algorithm using SCIAMACHY L1 v8 dataset. Left column shows validation results for the years 2003-2006 and right column shows the same for years 2007-2011. Solid lines are median differences of retrieved profiles and, dashed lines are median differences with a priori profiles.



**Table 5.** Validated level-2 product statistics

| Year | # (SH, Tr, NH) | dfs | n_iter | sza [°] | Stratosphere [1000-100] hPa $\left\langle \frac{a_{\text{opera}} - a_{\text{sonde}}}{a_{\text{sonde}}} \right\rangle$ [%] (SH, Tr, NH) | Troposphere [100-10] hPa $\left\langle \frac{a_{\text{opera}} - a_{\text{sonde}}}{a_{\text{sonde}}} \right\rangle$ [%] (SH, Tr, NH) |
|---|---|---|---|---|---|---|
| 2003 | (32, 107, 260) | 4.2 ± 0.4 | 5.0 ± 2.4 | 47.3 ± 16.3 | (-5.0, -1.6, +0.8) | (**+12.4**, **+40.4**, +2.5) |
| 2004 | (80, 121, 439) | 4.3 ± 0.5 | 5.0 ± 2.1 | 50.6 ± 18.3 | (-3.9, -1.3 , +0.1) | ( +5.9, **+27.8**, +0.1) |
| 2005 | (99, 121, 397) | 4.3 ± 0.5 | 5.0 ± 2.2 | 58.2 ± 19.0 | (-3.9, +1.3, +1.5) | ( +2.9, **+21.3**, -3.6) |
| 2006 | (78, 147, 481) | 4.1 ± 0.5 | 5.0 ± 2.2 | 45.8 ± 18.9 | (-1.4, +2.1, +2.8) | ( +5.3, **+17.4**, -9.7) |
| 2007 | (78, 88, 390) | 4.1 ± 0.5 | 5.0 ± 2.5 | 59.9 ± 19.5 | (+0.4, +4.2, +4.0) | ( -4.6, +0.7, -11.2) |
| 2008 | (119, 119, 373) | 4.2 ± 0.5 | 5.0 ± 2.0 | 54.2 ± 19.5 | (-0.3, +5.7, +6.4) | ( +3.9, -4.5, -12.0) |
| 2009 | (81, 101, 315) | 3.8 ± 0.5 | 5.0 ± 2.4 | 54.1 ± 19.0 | (+1.3, +11.6, +6.3) | ( -5.8, **-37.8**, **-22.9**) |
| 2010 | (90, 76, 360) | 3.8 ± 0.5 | 5.0 ± 2.4 | 56.3 ± 19.1 | (+3.0, +11.8, +6.5) | (-12.1, **-40.1**, **-22.3**) |
| 2011 | (96, 60, 327) | 3.8 ± 0.5 | 5.0 ± 2.5 | 58.8 ± 19.9 | (+1.0, +11.6, +8.4) | (-12.0, **-47.2**, **-22.8**) |

*Column 1,3,4*: Same as in Table 3, *Column 2*: Same as in Table 3 except number of pixels are given for each latitude zone: Southern Hemisphere (SH), Tropics (Tr), Northern Hemisphere (NH), *Column 5*: Median solar azimuth angle in degrees ± standard deviation for all pixels in that row, *Column 6* Medians of relative difference in profiles and sondes for stratosphere for SH, Tr and NH, *Column 7*: Same as in *Column 6* for troposphere

**Table 6.** Antarctic Ozone nadir column statistics

| Year | # [45°S:55°S] | # [55°S:70°S] | # [70°S:90°S] | [45°S:55°S] $\left\langle \sigma_{O3}^{\text{top}} \right\rangle$ | [55°S:70°S] $\left\langle \sigma_{O3}^{\text{mid}} \right\rangle$ | [70°S:90°S] $\left\langle \sigma_{O3}^{\text{bot}} \right\rangle$ |
|---|---|---|---|---|---|---|
| 2003 | [2 30] | [5 51] | [2 92] | 2.68 ± 0.30 | 2.72 ± 0.48 | 2.74 ± 0.90 |
| 2004 | [4 28] | [12 56] | [2 101] | 2.74 ± 0.29 | 2.76 ± 0.27 | 2.65 ± 0.54 |
| 2005 | [4 28] | [16 60] | [1 106] | 2.78 ± 0.32 | 2.77 ± 0.32 | 2.72 ± 0.54 |
| 2006 | [2 30] | [4 60] | [1 85] | 3.00 ± 0.46 | 3.05 ± 0.35 | 3.23 ± 0.63 |
| 2007 | [1 30] | [3 60] | [3 60] | 2.67 ± 0.47 | 2.82 ± 0.42 | 2.87 ± 0.44 |
| 2008 | [2 25] | [4 45] | [1 68] | 3.05 ± 0.45 | 3.03 ± 0.37 | 3.05 ± 0.40 |
| 2009 | [2 29] | [1 60] | [1 106] | 3.16 ± 0.56 | 3.04 ± 0.37 | 2.90 ± 0.61 |
| 2010 | [1 29] | [4 53] | [1 66] | 2.30 ± 0.64 | 3.26 ± 0.42 | 3.06 ± 0.42 |
| 2011 | [5 29] | [3 56] | [1 110] | 3.14 ± 0.58 | 3.30 ± 0.39 | 3.30 ± 0.47 |

*Column 2*:[Minimum Maximum] # SCIAMACHY pixels used in averaging ozone column amounts for the top latitude zone (see text), *Column 3,4*: Same as *Column 2* for middle and bottom latitude zones (see text), *Column 5*: Median uncertainty in stratospheric ozone column for top latitude zone ± standard deviation, *Column 6,7*: Same as *Column 5* for middle and bottom latitude zones





**Figure 8.** Vertically integrated ozone columns (in DU) for the years 2003-2006 in left column for latitude bands $45°$S:$55°$S (top row), $55°$S:$70°$S (middle row), $70°$S:$90°$S (bottom row). Each circle is the minimum value of daily retrieved quantity. The same is shown for years 2007-2011 in the right column. The time series are shown for months from 1 August - 1 Dec as labelled in x-axis where a is August, s is September, o is October, n is November and d is December. Each dot is a median of the total column amount for that day corresponding to the latitude bands and its size represents the number of pixels (states) used in computing the median.





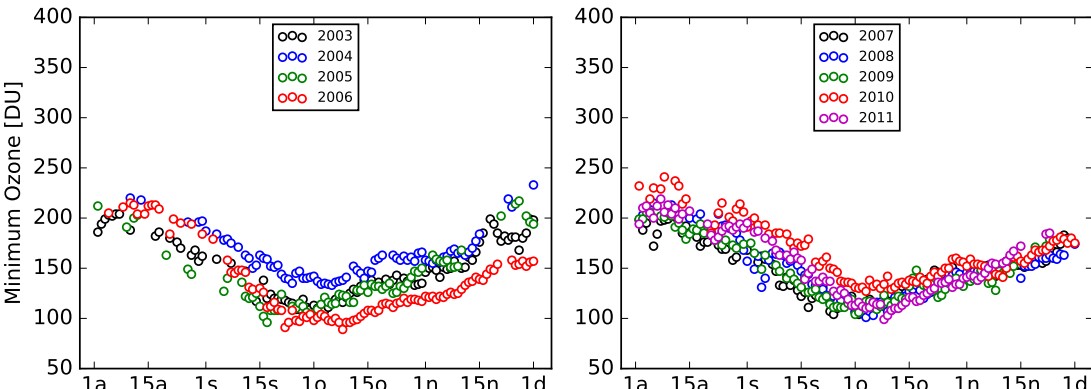

**Figure 9.** Same as in Figure 8 using assimilated MSR v2 data (see text). Daily minimum ozone is plotted for latitude band $70°$S:$90°$S

**Table 7.** Statistics of SCIAMACHY nadir ozone profile retrievals for the Antarctic region.

| Year | # Top | # Mid | # Bot | Top | Mid | Bot | $\langle \sigma_{O3}^{\mathrm{top}} \rangle$ | $\langle \sigma_{O3}^{\mathrm{mid}} \rangle$ | $\langle \sigma_{O3}^{\mathrm{bot}} \rangle$ |
|------|-------|-------|-------|------|------|------|------|------|------|
| 2003 | 1656 | 3514 | 3784 | 5 / 530.9 | 7.0/616.4 | 10.0/5750.2 | 0.55±0.82 | 0.59±1.07 | 0.44±1.18 |
| 2004 | 1777 | 4011 | 4845 | 5 / 556.6 | 6.0/616.5 | 10.0/660.1 | 0.56±0.83 | 0.62±1.07 | 0.46±1.18 |
| 2005 | 1877 | 3938 | 5103 | 5 / 579.2 | 6.0/647.7 | 10.0/713.5 | 0.56±0.84 | 0.62±1.07 | 0.46±1.22 |
| 2006 | 1727 | 3406 | 3278 | 5 / 559.2 | 6.0/652.8 | 10.0/1450.1 | 0.56±0.84 | 0.63±1.06 | 0.49±1.24 |
| 2007 | 1445 | 3141 | 2501 | 4 / 638.6 | 6.0/716.3 | 10.0/788.0 | 0.57±0.86 | 0.64±1.06 | 0.48±1.16 |
| 2008 | 1244 | 2629 | 2772 | 4 / 586.7 | 6.0/697.8 | 10.0/1672.0 | 0.57±0.84 | 0.63±1.09 | 0.48±1.23 |
| 2009 | 1664 | 3576 | 4019 | 4 / 645.2 | 6.0/753.0 | 10.0/1560.0 | 0.57±0.85 | 0.64±1.09 | 0.49±1.26 |
| 2010 | 1005 | 2071 | 2207 | 5 / 706.8 | 6.0/779.4 | 10.0/1826.8 | 0.56±0.86 | 0.65±1.11 | 0.49±1.27 |
| 2011 | 1776 | 3657 | 4481 | 4 / 757.6 | 6.0/786.0 | 10.0/3591.4 | 0.56±0.85 | 0.64±1.10 | 0.47±1.28 |

*Column 2,3,4*: # pixels used in computing median quantities with top: $45°$S:$55°$S, middle: $55°$S:$70°$S, bottom: $70°$S:$90°$S, *Column 5,6,7*: median `n_iter`/median cost function for top, mid and bot respectively, *Column 8,9,10*: median relative uncertainty in ozone layer per height $\pm$ st. dev. for top, mid and bot respectively





**Figure 10.** Nadir ozone vertical profile medians shown in Dobson Units per layer for years 2003-2006 in the left column for latitude bands 45°S:55°S (top row), 55°S:70°S (middle row), 70°S:90° (bottom row). The same is shown for years 2007-2011 in the right column.



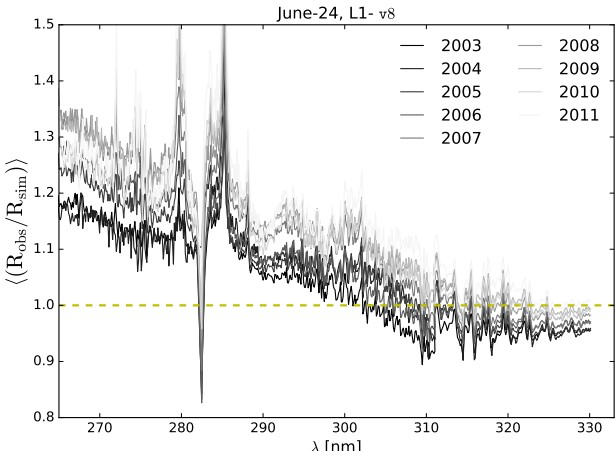

**Figure 11.** Mean ratio of observed ($R_{meas}$) to simulated model spectra ($R_{sim}$) for a day of observations for each year. The fading black colour goes from 2003 to 2011. A horizontal line where the ratio is one is shown for reference of the deviation of the ratio. Please note the odd behaviour of the spectral ratio around 283 nm which was found to be due to a jump in the spectral value of the Earth radiance between transitions of clusters 3 and 4 owing to the difference in the integration time between them. This region is blocked in our retrieval algorithm (See Sect. 2).

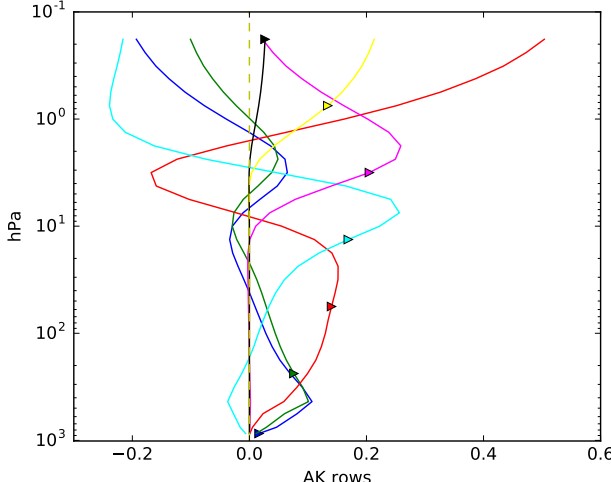

**Figure A1.** Example of the averaging kernel (AK) shape for a subset of seven layers from a 32-layer SCIAMACHY ozone profile retrieval. The triangles are the nominal altitudes of the retrieval layers. This subset of seven retrieval layers was selected uniformly ranging from top to bottom for clarity. From these AK curves it is clear that information on the ozone amount in a certain layer is also coming from other heights.



**Figure A2.** Comparison of validation of SCIAMACHY ozone profiles using ozone sondes with and without applying averaging kernels (AK) for the dataset v8. The solid lines indicate the use of the AK whereas the dashed lines indicate cases without AK. (cf. Fig. 6)