# Peer review of "Evaluation of SCIAMACHY Level-1 data versions using nadir ozone profile retrievals in the period 2003-2011"

_Atmospheric Measurement Techniques, 2017_

## Referee Comment (RC1) · Anonymous Referee #1 · 4 Jul 2017

I. General impression:

The authors present an application of KNMI's OPERA nadir ozone profile retrieval algorithm to three versions of SCIAMACHY IPP L1 data. Although the manuscript provides interesting and promising studies of the L1 versions and the resulting L2 retrieval, the work in general appears too broad and too superficial to be clear and satisfactory. It is therefore recommended to reconsider this manuscript after major revisions have been made. At least the extent of the result discussions (possibly with narrower focus) and scientific formulations (following a clear overall framework) have to be improved.

II. Specific comments:

[Figure]

The coverage of this manuscript is too broad to provide clear and satisfactory discussions: Topics addressed are L1 version intercomparison, instrument slit function retrieval, ozonesonde validation, satellite intercomparison (e.g. GOME-2), stratospheric, tropospheric, and total ozone (despite different claims, see below), Antarctic ozone depletion, averaging kernel shape and smoothing. . . As a result, the manuscript's title is unsatisfactory for its content, and vice-versa. Narrowing down the focus of the text would allow for appropriate coverage of these promising subjects and could strongly improve its scientific quality.

The overall readability of the manuscript could be (strongly) improved: - Word order mixing - Lack of use of definite and indefinite articles - It is difficult to derive the number of L1 versions that is actually studied from abstract and introduction, and to which of those the instrument slit function retrieval is applied. - Lack of a clear storytelling framework. - Untidy scientific formulations: "each nadir state is an area on the Earth's surface" (page 3, line 19-20), mixing of "degradation" and "degradation correction" which are quite different, "collocated geolocations indicating the location" (Figure 5 caption), "the median values of the quantity in the table conversions" (page 11 line 3), "[AK smoothing] gives a smoothed sonde profile which is more suitable to compare with the profile retrieved from the satellite instrument and influences the results" (page 11, lines 11-12), Table captions are typically too brief to be clear. . .

Page 6, line 14-15: "An inversion [. . .] is carried out until convergence is reached or until the maximum number of iterations is reached." What is the effect of the difference between the two options on the quality of the retrieval outcome?

Page 7-9, Section 3: The instrument slit function calibration parameters are well-explained, but their application is not: Is the slit function retrieval part of the OPERA retrieval? Why is it not applied to L1 v7 data to compare directly with v7_mfac? Why is their no squeeze in Channel 1? How can the retrieval start from 260 nm, if only data starting from 265 nm are considered? Why is the Channel distinction different in the text (at 308 nm) and in Table 1 (314 nm)? Please clarify.

Figure 4: Why show results for a narrow tropical band if the later focus is on the Antarctic ozone hole?

Starting from Section 5, reference is made to the "ESA CCI" requirements (including reference to a non-existent Section 4.3 on page 13, line 5). From the website link (page 12, line 17) and paper content, it is understood that the Ozone-CCI project is intended, but any clarification on the project or motivation for its reference is lacking.

The comparison with GOME-2 results obtained by van Peet et al. (2014) in the last paragraph of Section 5.3 seems to be of marginal support for the general discussion.

The added value of the discussion in Section 7 is poor. This section could be redistributed over the previous sections the keep relevant information together.

The conclusions in Section 8 are unsatisfactory in comparison with the amount of topics that have been covered in the manuscript and do not provide a summative overview that is clear to an external reader. Throughout the text, plot and table descriptions are not in balance with (too) short result discussions.

It is agreed with the associate editor that a discussion of the nadir ozone profile retrieval's averaging kernels is of importance for understanding the retrieved product. The single-pixel example (without discussion) and single-sentence conclusion on the effect of smoothing provided in Appendix A however are unsatisfactory: "The validation results are clearly less noisy and smoother for the case where the AK was applied to the ozone sondes." A lot more information on the retrieval performance can be derived from the averaging kernels.

III. Technical corrections:

Abstract, line 7: "focus on stratospheric ozone" and page 3, line 6 "we will focus exclusively on the study of ozone in the stratospheric region" does not match the many tropospheric and total column discussions appearing in the text.

Page 2, line 11: "their geographical coverage is limited to approximately 300 stations

worldwide" is exaggerated. It's rather about half of that.

Page 5, line 3-4: The statement on L1c data reproduction comes out of thin air. Please clarify.

Page 5, line 8: The (van Soest et al., 2005) reference for L1 v7.04–W data release in 2012 seems unfitting.

Figure 2: dashed lines, mean or median, explain order indication, "residuals for ∼300 spectra" in contrast with "each curve is a residual for one solar spectrum" in text (page 8, line 18)

Page 6, line 6-7: Provide proper definition and explanation for DFS. The current phrasing is vague.

Page 10, lines 5-6: The text provides conflicting information on the years that are covered in this section's discussion.

Page 12, line 6: Interpercentiles are indicated as errors, but in fact are (random) uncertainties on the relative differences.

Page 12, line 33: "solar azimuth angle" should be "solar zenith angle"

Section 6: Table numbers seem to be wrong.

Journal names in the References section are sometimes wrong (ACP and AMT are mixed) and both abbreviations and full names are used.

---

## Referee Comment (RC2) · Anonymous Referee #2 · 4 Jul 2017

General comments:

The authors retrieved nadir ozone profiles from SCIAMACHY, validated their results with ozonesondes and applied their data in the study of ozone hole in the Antarctic region during 2003-2011. This paper is ambitious to include retrievals (including slit function), validations and applications of SCIAMACHY's nadir ozone profiles. However, the focus of this paper is too broad, and unfortunately, many aspects were not well described and/or discussed. I think this paper could be separated into two papers at least: one validation paper and one application paper. In addition, the presentation of this paper could be significantly improved. Therefore, this paper needs a major revision

to meet AMT's publication requirements.

This paper simply used 1000-100 hPa and 100-10 hPa to separate the stratosphere and troposphere. Please use the tropopause pressures. Because tropopause pressures have strong temporal and spatial variations, a simple 100hPa separation could lead many scientific errors.

In Section 3, does OPERA have a slit function in the retrievals? Or did you apply a new slit function rather than the original slit function that improve the retrieval performance?

In Section 5, it is a clever idea to show the validation results in different latitude bands, as 90N-30N, 30N-30S and 30S-90S. However, it would be better to separate 90N-30N and 30S-90S latitude bands to 90N- 60N, 60N -30N and 30S-60S and 60S-90S. This is because satellite retrievals have a deficient performance at high latitude regions due to the large solar zenith angles. In the midlatitude, satellite retrievals usually are good.

Figure and table captions are too brief to understand.

Appendix A needs more discussions. "The validation results are clearly less noisy and smoother for the case where the AK was applied to the ozone sondes." Why?

Specific comments:

Page 1 Line 24: Tropospheric ozone could also come from the stratosphere.

Page 2 Line 7: "... the ozone trend layer...". Did you mean ozone layer trend?

Page 2 Line 9 – 12: Please add references to the descriptions and discussions of ozonesondes. Such as:

Deshler, T., Mercer, J. L., Smit, H. G. J., Stubi, R., Levrat, G., Johnson, B. J., Oltmans, S. J., Kivi, R., Thompson, A. M., Witte, J., Davies, J., Schmidlin, F. J., Brothers, G., and Sasaki, T.: Atmospheric comparison of electrochemical cell ozonesondes from different manufacturers, and with different cathode solution strengths: The Balloon Experiment on Standards for Ozonesondes, J. Geophys. Res., 113, doi: 10.1029/2007JD008975,

2008.

Smit, H. G. J., Straeter, W., Johnson, B. J., Oltmans, S. J., Davies, J., Tarasick, D. W., Hoegger, B., Stubi, R., Schmidlin, F. J., Northam, T., Thompson, A. M., Witte, J. C., Boyd, I., and Posny, F.: Assessment of the performance of ECC-ozonesondes under quasi-flight conditions in the environmental simulation chamber: Insights from the Juelich Ozone Sonde Intercomparison Experiment (JOSIE), J. Geophys. Res., 112, 19306, 2007.

Page 2 Line 16: Please add references for IASI and TES.

Page 2 Line 34: In your abstract, you mentioned only two version of SCIAMACHY, v7 and v8. Please specify three version numbers here.

Page 3 Line 2: Similarly, please specify the version number here.

Page 3 Line 4-6: The focus of this study is to apply the retrieval algorithm in different SCIAMACHY L1 data with validation against global ozonesonde observations during 2003-2011, and analyze the stratospheric ozone in the Antarctic region. Please rephrase the focus part.

Page 3 Line 7: It is the first time you use OPERA in your text, except the abstract. Please specify the full name here.

Page 5 Line 9: The link to describe goes to V8mfac, please revise the link.

Page 5 Line 17: "difference between this version with the ones above...". Please change with to and.

Page 5 Line 24: It would be a good idea to have a paragraph to describe briefly the retrieval theory in Rogers (2000), like van Peet et al. (2014) did.

van Peet, J. C. A., van der A, R. J., Tuinder, O. N. E., Wolfram, E., Salvador, J., Levelt, P. F., and Kelder, H. M.: Ozone ProfilE Retrieval Algorithm (OPERA) for nadir-looking satellite instruments in the UV–VIS, Atmos. Meas. Tech., 7, 859-876, doi: 10.5194/amt-

7-859-2014, 2014.

Page 5 Line 25: "... using UV and Visible wavelengths." Did you mean UV wavelengths here?

Page 10 Section 4.1:

1. I think it is a good idea to also include satellite retrieved ozone profiles 2002 and 2012 in your validation, although you have the entire year data.

2. Why did you only present mean profiles in 2003 and 2009? Could you list annually mean retrieved ozone profiles from 2002-2012 (or 2003-2011)? It would indicate the temporal variation of ozone profiles based on different level 1 data, and the instrument degradation. Page 11 Line 4: The ozonesonde ozone profiles have different top pressures because of some early burst balloons. Did you apply correction factors to ozonesonde profiles? Some ozone data processes should be described here.

Page 11 Line 10: please specify the equation of average kernel convolution. And Why did you need to convolve ozonesonde profiles with SCIAMACHY average kernels.

Page 10 Line 29: "... the lower- middle stratosphere (100 – 10 hPa)...". The lower-mid stratosphere is roughly 100-10 hPa. But it is key to point calculate the tropopause pressures. I suggest including yearly mean tropopause pressure here.

Page 11 line 10: "An ideal agreement between sonde and satellite would give a difference of zero at all layer heights." This is not necessarily true. Sometimes, small biases are because of retrieval information coming from the a priori due to the low satellite sensitivity in the lower troposphere,

Figure 3. Please convert x-axis ticks from year 0-8 to year 2002-2010 or 02 – 10.

Figure 4: In the right columns, please indicate that they are mean ozone profiles based on different level 1 data within the 10N and 10S region. And what do these dashed lines mean?

Figure 5: Please add latitude and longitude ticks in the figures.

Figure 6: What kind of uncertainty did you plot here? Standard deviations?

[Figure]

---

## Referee Comment (RC3) · Anonymous Referee #3 · 5 Jul 2017

The manuscript "Nadir ozone profile retrieval from SCIAMACHY and its application to the Antarctic ozone hole in the period 2003–2011" by S. Shah et al. presents an evaluation of several versions of Level1 SHIAMACHY nadir data that is done by validating Level 2 ozone profile retrievals. The approach for Level 1 evaluation presented in this paper might be promising, but it requires a deeper analysis in order to connect observed changes in the ozone profile retrievals to a specific calibration/adjustment applied to Level 1 data, which I believe is not completed in this study. The second portion of the paper is dedicated to a validation of the ozone profiles derived using latest Level 1 dataset (v8) against sonde observations. These ozone retrievals are also employed to look for inter-annual ozone variability over Antarctica. Unfortunately, the paper is

not very well focused and covers a large number of topics, while none of them is fully investigated. The title of the paper, the abstract and conclusions do not reflect the content of the manuscript. This manuscript needs a major revision before publishing in the AMT.

General comments:

- The focus of the paper should be narrowed down. If the focus of the study is on evaluation of different Level1 data, than the part with the Antarctic ozone hole should be removed from the paper, the title needs to reflect the goal of the study, and appropriate changes should be made in the abstract and conclusions.

- In Introduction (page 2) authors presented a very "skewed" overview of the existing satellite methods of retrieving ozone profiles. Ozone profile retrievals derived from nadir SBUV/SBUV-2 sensors span for more than 40 years. There is also a number of limb ozone profile datasets, besides described SCHIAMACHY limb, with high vertical resolution, like MLS, MIPAS, OSIRIS and OMPS-LP, as well as from occultation instruments like ACE-FTS, GOMOS and SAGE II. Some of these datasets overlap with SCHIAMACHY mission and can be used for validation in addition to sonde data.

- Section 3: I do not understand what had been done with the Slit Function in this study. Have you applied the SF corrections described in section 3 to Level 1 data before doing inversion? Did you apply these SF corrections to all 3 versions of Level 1 data? Or do you provide description of the SF corrections that had already been implemented in Level 1 data? Please, clarify that in the text.

- Figures 6 and 7 show differences with sondes for early and later years of SCHIMACHY mission. It is obvious from these figures that the vertical pattern of differences has changed significantly (not just the absolute differences) over the instrument lifetime. It is not reflected in the discussion, however it is very important for any scientific application as it will lead to wrong conclusions. Obviously, such a change in the vertical pattern of differences points to a significant drift in O3 retrievals. Thus, you can

not draw any reliable conclusions about inter-annual ozone variability over Antarctica using these O3 dataset. If the goal of the paper to look on effect of Level 1 adjustments, then it is worth to look at O3 time series at several levels to determine time-dependent changes and connect them to a drift in different spectral channels.

- Sections 4-6: All captions for figures and tables should be revisited and all data/lines should be clearly explained (see more below in "specific comments"). Many conclusions presented in these sections are not obvious for readers and more explanations and evidences are needed.

- Section 7 seems to be disconnected with the previous discussion. It is not clear what is the purpose of this section.

- It was mentioned several times in the text and in the abstract that the focus will be on the stratospheric ozone. And this would be a reasonable approach, since nadir sensors are not expected to produce high quality ozone profiles in the troposphere (especially relative to sonde measurements). It is not clear to me why results below 200-300 hPa are shown in the figures. Also several tables have extra lines to show statistics for the tropospheric values. It concerns me that some of the conclusions regarding to performances of different versions are based on the tropospheric results rather than stratospheric.

Specific comments:

- Section 2.2, page 6, lines 13-15: Since the vertical resolution of nadir measurements are limited they are sensitive to the a priori profiles as well as to assumed a priori and measurement covariance matrix. Please, specify which a priori data were used in the study, because Table 2 says that 3 different a priori data sets available in OPERA. Also, please, explain how matrices required for the Optimal Estimation were set in your study. Do Level 1 SHIAMACHY data come with the uncertainties that you use in the retrieval algorithm or do you have to assume these uncertainties? Are these measurement uncertainties the same for all Level 1 data sets that you tested here?

- Figure 4. Please, specify in the caption what horizontal dashed lines represent. Are they shown here to make a connection between specific spectral ranges in radiances to altitude range in O3 profiles? It's also would be helpful to add the mean/median equatorial sonde profiles for 2003 and 2009 as a reference.

-Sections 4-6: Median values for biases and other characteristics are reported instead of mean values. Do you have a specific reason for using median values? Have you found many outliers or do you believe that the distribution of differences is mostly skewed to a particular direction, so the mean values are not representative? Please, explain that in the text.

- Section 5.2, lines 1-10: This part of the text is very confusing. Do you make a conclusion "This suggests that the quality of nadir L1 data is still poor" based on the fact that differences in tropospheric ozone are too large? This part needs to be revised.

- Appendix. The Averaging Kernels for nadir observations change significantly with latitude and season. Showing just one example of the AK is not sufficient. I would suggest to show DFS profiles for different latitude bands and seasons, and specifically for the Antarctic latitudes in Sep-Oct. I also believe that this discussion belongs to section 2.2 where you should describe the main characteristics of the SCHIAMACHY nadir ozone retrievals.

- Section 5.1: On my opinion authors didn't provide enough evidences and explanations to demonstrate that version v8 is any better than v7_mfac.

- Section 5.2: I believe that results presented in Fig. 7 and Table 5 are not enough to call this section 'Validation of v8'. For instance in section 7, authors speculate that on the days when the instrument was heated the measured radiances were affected. Have you tried to isolate and remove those days from your analysis? Do you see improvements/changes in the results?

- Section 6: The analysis shown in this section is insufficient. There are many total

ozone observations available for the considered time period that can be used to validate integrated ozone columns instead of looking at the reanalysis data. Also, the statement in the conclusion "we investigated the Antarctic ozone profile behavior in the austral spring season" doesn't correspond to the work shown in Section 6. There are many satellite ozone observations that overlap (or partially overlap) with SCHIA-MACHY mission like SBUV/2 NOAA-17, Odin OSIRIS, ACE-FTS, Aura MLS and MI-PAS. Comparisons with these correlative measurements would help you to understand how well SCHIAMACHY nadir profiles can describe the vertical ozone distribution inside the ozone hole, and therefore if this dataset is suitable for studying inter-annual ozone variability over Antarctica. Without this extensive analysis it is not possible to claim that SCHIAMACHY nadir ozone profiles can be applied for the scientific analysis.

Minor correction/typos:

- Section 6 p 14 line 35 and page 15 lines 1-2: I don't see the cost function or # of iterations (that are not shown in Table 6!!!). Do you mean Table 7 here?

- page 5, line 29: Should be "This allows" instead of "This amounts"

- Table 2, Pressure grid- it would be useful to see the pressure grid used in your retrieval algorithm.

- Table 3. Please, in the Table caption make a connection to the corresponding Figure 4. Also, it's not clear what do you mean by # of pixels? Is it a number of profiles considered for this comparison? Why is it different for different versions of Level 1 data? Please, spell "n_inter" as "number of iterations" in the caption. Could you explain what does it mean "median n_iter for #pixels"? Do you mean median number of iterations for the considered pixels (Profiles)? It should be "Column 7: standard deviation of Column 6".

- Tables 4 and 5. Please, fix "Troposphere [1000-100] hPa" and "Stratosphere [100-10] hPa". The values shown in columns 6 and 7: are these biases for integrated

stratospheric/tropospheric columns or mean over specified altitude range?

- Table 6: Please, clarify what quantities are shown in columns 2-4. Do they show min and max numbers of O3 profiles used to calculate the daily zonal mean value? It says "Column 5: Median uncertainty in stratospheric ozone column". Is it a correct label? What did you use to determine the tropopause pressures? Also why did you show total ozone values on Figure 8 and not stratospheric columns?

- Table 7. Please, explain what is "relative uncertainties in ozone layer per height". Section 6, Figure 8 and Table 6 show results for total ozone columns. The numbers for sigma shown in Tables 6 and 7 are not the same. Please, add clear explanation of results shown in these tables.

---

## Author Comment (AC1) · 24 Nov 2017

**Response to Anonymous Referee #1**

We thank the referee for the constructive comments and address them point-by-point. The comment is copied below in normal font, and the response is in italics.

The paper has been re-organized in order to make the focus on SCIAMACHY L1 evaluation clearer: the part on the Antarctic ozone hole has been removed from the paper, and the Appendix has been incorporated into the main text. A new figure showing profiles of the DFS for all seasons and two representative years has been included.

[Figure]

The text has been revised throughout the paper. The textual changes are indicated in colored text in the manuscript version showing differences.

**Specific comments**

1. The coverage of this manuscript is too broad to provide clear and satisfactory discussions: Topics addressed are L1 version intercomparison, instrument slit function retrieval, ozonesonde validation, satellite intercomparison (e.g. GOME-2), stratospheric, tropospheric, and total ozone (despite different claims, see below), Antarctic ozone depletion, averaging kernel shape and smoothing ... As a result, the manuscript's title is unsatisfactory for its content, and vice-versa. Narrowing down the focus of the text would allow for appropriate coverage of these promising subjects and could strongly improve its scientific quality. The overall readability of the manuscript could be (strongly) improved: - Word order mixing - Lack of use of definite and indefinite articles - It is difficult to derive the number of L1 versions that is actually studied from abstract and introduction, and to which of those the instrument slit function retrieval is applied. - Lack of a clear storytelling framework. - Untidy scientific formulations: "each nadir state is an area on the Earth's surface" (page 3, line 19-20), mixing of "degradation" and "degradation correction" which are quite different, "collocated geolocations indicating the location" (Figure 5 caption), "the median values of the quantity in the table conversions" (page 11 line 3), "[AK smoothing] gives a smoothed sonde profile which is more suitable to compare with the profile retrieved from the satellite instrument and influences the results" (page 11, lines 11-12), Table captions are typically too brief to be clear ...

   *Author Response:*
   *We have narrowed down the focus of the article to the SCIAMACHY L1 version evaluation, through intercomparison of L2 retrievals and L2 validation. Slit func-*

*tion retrieval has been moved to the Discussion as it is not used in the version comparison but is still a part of the calibration of the L1 data. The title has been changed appropriately. We have extensively adjusted the text for better formulation and readability; the many changes are indicated in colored text.*

2. Page 6, line 14-15: "An inversion [...] is carried out until convergence is reached or until the maximum number of iterations is reached." What is the effect of the difference between the two options on the quality of the retrieval outcome?

   *Author Response:*
   *Convergence is defined as when the difference between the cost function between two iterative solutions are less than a few percent. This typically happens before 10 iterations. If it exceeds 10 iterations, the retrieval is considered unsuccessful. This is now clarified in the text; see also Table 2.*

3. Page 7-9, Section 3: The instrument slit function calibration parameters are well-explained, but their application is not: Is the slit function retrieval part of the OPERA retrieval? Why is it not applied to L1 v7 data to compare directly with v7_mfac? Why is their no squeeze in Channel 1? How can the retrieval start from 260 nm, if only data starting from 265 nm are considered? Why is the Channel distinction different in the text (at 308 nm) and in Table 1 (314 nm)? Please clarify.

   *Author Response:*
   *We have moved the Slit function section to the Discussion section in the new organization of the paper. Slit function retrieval is not a part of the OPERA retrieval used in this paper. The reason is that the slit function correction is of the order of 10 times smaller than the degradation correction, so slit function correction currently does not contribute much to the improvement of the data quality; this is now explained in the Discussion. There is no SF squeeze in channel 1, because adding squeeze does not improve the slit function retrieval there. The retrieval indeed starts at 265 nm. The division of 308 nm corresponds only to*

*the slit function retrieval. SF retrieval on Channel 1 from 265-308 nm gives the lowest residuals rather than taking 265-314 nm. Taking the 308-330 nm spectra in Channel 2 gives the lowest residuals. This is now explained in the Slit function part in the Discussion section.*

4. Figure 4: Why show results for a narrow tropical band if the later focus is on the Antarctic ozone hole?

   *Author Response:*
   *We have removed the Antarctic ozone hole analysis from the paper, as the focus of this paper is the L1 data quality analysis.*

5. Starting from Section 5, reference is made to the "ESA CCI" requirements (including reference to a non-existent Section 4.3 on page 13, line 5). From the website link (page 12, line 17) and paper content, it is understood that the Ozone-CCI project is intended, but any clarification on the project or motivation for its reference is lacking. The comparison with GOME-2 results obtained by van Peet et al. (2014) in the last paragraph of Section 5.3 seems to be of marginal support for the general discussion. The added value of the discussion in Section 7 is poor. This section could be redistributed over the previous sections the keep relevant information together.

   *Author Response:*
   *We have corrected "ESA CCI", with which we mean the ESA Ozone-CCI project. From this project we take the required accuracy requirements for satellite ozone profile measurements. We corrected the section reference. The Discussion section (now Sec. 5) has been expanded to include two topics related to L1 data calibration: degradation and slit function retrieval and their relative importance in retrieving L2 ozone profiles. We have removed the discussion about the Antarctic ozone, so the text is only pertaining to L1 data evaluation.*

6. It is agreed with the associate editor that a discussion of the nadir ozone profile retrieval's averaging kernels is of importance for understanding the retrieved product. The single-pixel example (without discussion) and single-sentence conclusion on the effect of smoothing provided in Appendix A however are unsatisfactory: "The validation results are clearly less noisy and smoother for the case where the AK was applied to the ozone sondes." A lot more information on the retrieval performance can be derived from the averaging kernels.

*Author Response:*
*We agree that the averaging kernel is an important concept in ozone profile retrieval. Therefore the AK is discussed in the retrieval algorithm part (Sec 2.2) with one graphical example (Fig. 2). In Sec. 4 we added more text on the application of the AK for validation, with one graphical example of its impact on the validation (Fig. 8). We have added a new figure, Fig. 4, showing the DFS profiles for different seasons for the years 2003 and 2009.*

**Technical corrections**

1. Abstract, line 7: "focus on stratospheric ozone" and page 3, line 6 "we will focus exclusively on the study of ozone in the stratospheric region" does not match the many tropospheric and total column discussions appearing in the text.

   *Author Response:*
   *We corrected the main text. Please note that due to reorganization of the paper the part on total columns has been removed.*

2. Page 2, line 11: "their geographical coverage is limited to approximately 300 stations worldwide" is exaggerated. It's rather about half of that.

   *Author Response:*
   *We corrected this in the Introduction, and added a reference to a publication on ozone networks.*
3. Page 5, line 3-4: The statement on L1c data reproduction comes out of thin air. Please clarify.

   *Author Response:*
   *We now clarify the L1c data reproduction in the beginning of Sec 2.1.*

4. Page 5, line 8: The (van Soest et al., 2005) reference for L1 v7.04–W data release in 2012 seems unfitting.

   *Author Response:*
   *We removed the reference.*

5. Figure 2: dashed lines, mean or median, explain order indication, "residuals for 300 spectra" in contrast with "each curve is a residual for one solar spectrum" in text (page 8, line 18)

   *Author Response:*
   *This is now Fig. 9. The meaning of dashed lines is now explained. We only show representative residuals for 300 spectra in the figure for clarity without overloading the size of the file. These 300 spectra are representative for the mission lifetime from 2002-2012. We explain this in the text belonging to the figure.*

6. Page 6, line 6-7: Provide proper definition and explanation for DFS. The current phrasing is vague.

   *Author Response:*
   *DFS is now explained in Sec. 2.2. In the new Fig. 4 the DFS is now shown for different seasons and years.*

7. Page 10, lines 5-6: The text provides conflicting information on the years that are covered in this section's discussion.

*Author Response:*
*We try to make it clear that the Slit function retrieval is done for 2002-2012, however the ozone validation and retrieval is done for 2003-2011 depending on the dataset version. The reasons are also explained.*

8. Page 12, line 6: Interpercentiles are indicated as errors, but in fact are (random) uncertainties on the relative differences.

   *Author Response:*
   *Agreed. Text is corrected in Sect. 4.1.*

9. Page 12, line 33: "solar azimuth angle" should be "solar zenith angle"

   *Author Response:*
   *Corrected.*

10. Section 6: Table numbers seem to be wrong.

    *Author Response:*
    *Corrected.*

11. Journal names in the References section are sometimes wrong (ACP and AMT are mixed) and both abbreviations and full names are used.

    *Author Response:*
    *Consistent journal names are now used in the References, and mistakes have been corrected.*

---

## Author Comment (AC2) · 24 Nov 2017

**Response to Anonymous Referee #2**

We thank the referee for the constructive comments and address them point-by-point. The comment is copied below in normal font, and the response is in italics.

The paper has been re-organized in order to make the focus on SCIAMACHY L1 evaluation clearer: the part on the Antarctic ozone hole has been removed from the paper, and the Appendix has been incorporated into the main text. A new figure showing profiles of the DFS for all seasons and two representative years has been included.

[Figure]

The textual changes are indicated in colored text in the manuscript version showing differences.

**General comments**

1. The authors retrieved nadir ozone profiles from SCIAMACHY, validated their results with ozonesondes and applied their data in the study of ozone hole in the Antarctic region during 2003-2011. This paper is ambitious to include retrievals (including slit function), validations and applications of SCIAMACHY's nadir ozone profiles. However, the focus of this paper is too broad, and unfortunately, many aspects were not well described and/or discussed. I think this paper could be separated into two papers at least: one validation paper and one application paper. In addition, the presentation of this paper could be significantly improved. Therefore, this paper needs a major revision to meet AMT's publication requirements.

   *Author Response:*
   *We have followed the suggestion of the reviewer: we have narrowed down the focus of the article to the evaluation of SCIAMACHY L1 data by validation of ozone profiles. We removed the application to the Antarctic ozone hole. We have furthermore drastically improved the text. The title has been changed accordingly. This was a major revision. See also response 1 to Referee #1.*

2. This paper simply used 1000-100 hPa and 100-10 hPa to separate the stratosphere and troposphere. Please use the tropopause pressures. Because tropopause pressures have strong temporal and spatial variations, a simple 100 hPa separation could lead many scientific errors.

   *Author Response:*
   *We decided that this improved strat/trop separation was not needed anymore,*

*since we changed the scope of the paper to L1 evaluation. We removed the analysis of the Antarctic ozone hole. In the validation part of the paper we still separate between 1000-100 and 100-10 hPa regions.*

3. In Section 3, does OPERA have a slit function in the retrievals? Or did you apply a new slit function rather than the original slit function that improve the retrieval performance?

   *Author Response:*
   *In Section 3: The OPERA retrievals of the paper use the default slit function provided with the SCIAMACHY L1 data. We did not apply the new slit function to the ozone profile retrievals. Please see response 3 to Referee #1.*

4. In Section 5, it is a clever idea to show the validation results in different latitude bands, as 90N-30N, 30N-30S and 30S-90S. However, it would be better to separate 90N-30N and 30S-90S latitude bands to 90N- 60N, 60N -30N and 30S-60S and 60S-90S. This is because satellite retrievals have a deficient performance at high latitude regions due to the large solar zenith angles. In the midlatitude, satellite retrievals usually are good.

   *Author Response:*
   *Agreed. We now separate the results for 90N-30N and 30S-90S latitude bands into 90N- 60N, 60N -30N and 30S-60S and 60S-90S bands. See Sect. 4 in the revised paper.*

5. Figure and table captions are too brief to understand. Appendix A needs more discussions. "The validation results are clearly less noisy and smoother for the case where the AK was applied to the ozone sondes." Why?

   *Author Response:*
   *Agreed. We have reworded and improved figure and table captions. We have included the appendix on the AK in the main text. The AK convolved validation*

[Figure]

*results are smoother, because the comparison takes into account the resolution of the satellite retrieval. This is explained at the end of Sect. 4.*

**Specific comments**

Please note that we do not comment on all Antarctic ozone analysis comments, since we removed this whole section and related text elsewhere in the paper from the revised version.

1. Page 1 Line 24: Tropospheric ozone could also come from the stratosphere.

   *Author Response:*
   *Agreed. Text has been corrected.*

2. Page 2 Line 7:". . . the ozone trend layer. . .". Did you mean ozone layer trend?

   *Author Response:*
   *Yes. Text corrected.*

3. Page 2 Line 9 – 12: Please add references to the descriptions and discussions of ozonesondes.

   *Author Response:*
   *We added the suggested references on ozonesondes.*

4. Page 2 Line 16: Please add references for IASI and TES.

   *Author Response:*
   *We added references on IASI and TES.*

5. Missing version numbers of L1 data on Page 3 Line 34 and on Page 3 Line 2.

*Author Response:*
*The version numbers are now given.*

6. Page 3 Line 7: It is the first time you use OPERA in your text, except the abstract. Please specify the full name here.

   *Author Response:*
   *Full name of OPERA is now specified in Sec. 2.2.*

7. Page 5 Line 9: The link to describe goes to V8mfac, please revise the link.

   *Author Response:*
   *We corrected the link.*

8. Page 5 Line 17: "difference between this version with the ones above: : :". Please change with to and.

   *Author Response:*
   *Text corrected.*

9. Page 5 Line 24: It would be a good idea to have a paragraph to describe briefly the retrieval theory in Rogers (2000), like van Peet et al. (2014) did. van Peet, J. C. A., van der A, R. J., Tuinder, O. N. E., Wolfram, E., Salvador, J., Levelt, P. F., and Kelder, H. M.: Ozone ProfilE Retrieval Algorithm (OPERA) for nadir-looking satellite instruments in the UV–VIS, Atmos. Meas. Tech., 7, 859-876, doi: 10.5194/amt-7-859-2014, 2014.

   *Author Response:*
   *Agreed. Description has been added to Sect. 2.2.*

10. Page 5 Line 25: ": : : using UV and Visible wavelengths." Did you mean UV wavelengths here?

    *Author Response:*
    *Yes, this is now corrected.*

11. Page 10 Section 4.1:
1. I think it is a good idea to also include satellite retrieved ozone profiles (for) 2002 and 2012 in your validation, although you have (not) the entire year data.
2. Why did you only present mean profiles in 2003 and 2009? Could you list annually mean retrieved ozone profiles from 2002-2012 (or 2003-2011)? It would indicate the temporal variation of ozone profiles based on different level 1 data, and the instrument degradation.

*Author Response:*
*(1) We do not include 2002 and 2012, because there are only a few months of data for those years which is not sufficient for validation.*
*(2) The two years 2003 and 2009 were chosen to show the effect of degradation. We did not make figures or tables for all the years, to not increase the size of the paper unnecessarily. The year-to-year variation of retrieved ozone profile quality is shown in Fig. 7.*

12. *Page 11 Line 4*: The ozonesonde ozone profiles have different top pressures because of some early burst balloons. Did you apply correction factors to ozonesonde profiles? Some ozone data processes should be described here.

*Author Response:*
*Correction factors to ozonesonde profiles are indeed applied. Ozone data processes are described a bit more, but the details are already described in van Peet et. al. 2015 to which we refer.*

13. Page 11 Line 10: Please specify the equation of average kernel convolution. And why did you need to convolve ozonesonde profiles with SCIAMACHY average kernels.

*Author Response:*
*We now better explain the role of averaging kernels. See also response 6 to Referee #1.*

14. Page 10 Line 29: "... the lower- middle stratosphere (100 – 10 hPa): : :". The lowermid stratosphere is roughly 100-10 hPa. But it is key to point calculate the tropopause pressures. I suggest including yearly mean tropopause pressure here.

*Author Response:*
*See our response to General Comment #2 above.*

15. Page 11 line 10: "An ideal agreement between sonde and satellite would give a difference of zero at all layer heights." This is not necessarily true. Sometimes, small biases are because of retrieval information coming from the a priori due to the low satellite sensitivity in the lower troposphere.

*Author Response:*
*Agreed. We removed the sentence.*

16. Figure 3. Please convert x-axis ticks from year 0-8 to year 2002-2010 or 02—10.

*Author Response:*
*We converted x-axis ticks from year 0-8 to 02—10.*

17. Figure 4: In the right columns, please indicate that they are mean ozone profiles based on different level 1 data within the 10N and 10S region. And what do these dashed lines mean?

*Author Response:*
*We have now clarified this.*

18. Figure 5: Please add latitude and longitude ticks in the figures.

*Author Response:*
*Added.*

19. Figure 6: What kind of uncertainty did you plot here? Standard deviations?

*Author Response:*
*The uncertainties plotted here are the 25-75 percentile differences.*

---

## Author Comment (AC3) · 24 Nov 2017

**Response to Anonymous Referee #3**

We thank the referee for the constructive comments and address them point-by-point. The comment is copied below in normal font, and the response is in italics.

The paper has been re-organized in order to make the focus on SCIAMACHY L1 evaluation clearer: the part on the Antarctic ozone hole has been removed from the paper, and the Appendix has been incorporated into the main text. A new figure showing profiles of the DFS for all seasons and two representative years has been included.

The text has been revised throughout the paper. The textual changes are indicated in colored text in the manuscript version showing differences.

**General comments**

1. The focus of the paper should be narrowed down. If the focus of the study is on evaluation of different Level1 data, than the part with the Antarctic ozone hole should be removed from the paper, the title needs to reflect the goal of the study, and appropriate changes should be made in the abstract and conclusions. In Introduction (page 2) authors presented a very "skewed" overview of the existing satellite methods of retrieving ozone profiles. Ozone profile retrievals derived from nadir SBUV/SBUV-2 sensors span for more than 40 years. There is also a number of limb ozone profile datasets, besides described SCHIAMACHY limb, with high vertical resolution, like MLS, MIPAS, OSIRIS and OMPS-LP, as well as from occultation instruments like ACE-FTS, GOMOS and SAGE II. Some of these datasets overlap with SCIAMACHY mission and can be used for validation in addition to sonde data.

   *Author Response:*
   *As requested by the reviewer we have narrowed down the focus of the article to an evaluation of different L1 datasets. The ozone hole part has been removed. The title has been changed accordingly. In the Introduction we added other types of satellite ozone profile measurements, but we do not aim for completeness, since our focus is SCIAMACHY.*

2. Section 3: I do not understand what had been done with the Slit Function in this study. Have you applied the SF corrections described in section 3 to Level 1 data before doing inversion? Did you apply these SF corrections to all 3 versions of Level 1 data? Or do you provide description of the SF corrections that had

already been implemented in Level 1 data? Please, clarify that in the text.

*Author Response:*
*We agree that this was not clear. We used the default slit function in the retrievals. We evaluated the slit function of SCIAMACHY to see if we can calibrate the solar spectra better. The answer turned out to be yes, however the calibration errors from insufficient degradation corrections are significantly worse than those from the slit function. Thus applying slit function fitting would not significantly change the results. This is now presented in a more organised way in the Discussion section.*

3. Figures 6 and 7 show differences with sondes for early and later years of SCIA-MACHY mission. It is obvious from these figures that the vertical pattern of differences has changed significantly (not just the absolute differences) over the instrument lifetime. It is not reflected in the discussion, however it is very important for any scientific application as it will lead to wrong conclusions. Obviously, such a change in the vertical pattern of differences points to a significant drift in O3 retrievals. Thus, you can not draw any reliable conclusions about inter-annual ozone variability over Antarctica using these O3 dataset. If the goal of the paper to look on effect of Level 1 adjustments, then it is worth to look at O3 time series at several levels to determine time-dependent changes and connect them to a drift in different spectral channels.

*Author Response:*
*We thank the reviewer for this important observation. Since we have omitted the application to the Antarctic ozone hole, this comment is not so relevant anymore for the current paper. We added this remark to the discussion of the figure with ozone profile validation for all years (now Fig. 7).*

4. Sections 4-6: All captions for figures and tables should be revisited and all data/lines should be clearly explained (see more below in "specific comments").

Many conclusions presented in these sections are not obvious for readers and more explanations and evidences are needed.

*Author Response:*
*We have revised all captions of figures and tables. The text of the paper has been revised to improve clarity.*

5. Section 7 seems to be disconnected with the previous discussion. It is not clear what is the purpose of this section.

*Author Response:*
*The discussion section is revised. It now contains the calibration issues for potential future improvement of the L1 data, which connects it with the rest of the paper which presents the evaluation of the current L1 datasets.*

6. It was mentioned several times in the text and in the abstract that the focus will be on the stratospheric ozone. And this would be a reasonable approach, since nadir sensors are not expected to produce high quality ozone profiles in the troposphere (especially relative to sonde measurements). It is not clear to me why results below 200-300 hPa are shown in the figures. Also several tables have extra lines to show statistics for the tropospheric values. It concerns me that some of the conclusions regarding to performances of different versions are based on the tropospheric results rather than stratospheric.

*Author Response:*
*We prefer to show the entire retrieved ozone profile, thus including the troposphere. However, we focus on stratospheric ozone, since the validation shows that the retrieval has a poor performance in the troposphere. Our conclusion is that the retrieval is most reliable in the stratosphere and we mention this more clearly in the paper in the conclusions and validation sections.*
**Specific comments**

1. Section 2.2, page 6, lines 13-15: Since the vertical resolution of nadir measurements are limited they are sensitive to the a priori profiles as well as to assumed a priori and measurement covariance matrix. Please, specify which a priori data were used in the study, because Table 2 says that 3 different a priori data sets available in OPERA. Also, please, explain how matrices required for the Optimal Estimation were set in your study. Do Level 1 SHIAMACHY data come with the uncertainties that you use in the retrieval algorithm or do you have to assume these uncertainties? Are these measurement uncertainties the same for all Level 1 data sets that you tested here?

   *Author Response:*
   *We used the McPeters et al., 2007 a-priori profiles; this is now mentioned in the text. We have used the measured reflectance variances and a relative noise-floor (systematic noise) of 0.15.*

2. Figure 4. Please, specify in the caption what horizontal dashed lines represent. Are they shown here to make a connection between specific spectral ranges in radiances to altitude range in O3 profiles? It's also would be helpful to add the mean/median equatorial sonde profiles for 2003 and 2009 as a reference.

   *Author Response:*
   *The horizontal lines are explained in the text. Note that this is Figure 3 in the revised paper. We prefer to keep the comparison with sondes to Sec. 4, because there we compare the sondes systematically with the SCIAMACHY profiles.*

3. Sections 4-6: Median values for biases and other characteristics are reported instead of mean values. Do you have a specific reason for using median values? Have you found many outliers or do you believe that the distribution of differences

is mostly skewed to a particular direction, so the mean values are not representative? Please, explain that in the text.

*Author Response:*
*We found a few outliers, but not many. However we choose medians because we consider them more representative than the means when using large datasets, which is the case here.*

4. Section 5.2, lines 1-10: This part of the text is very confusing. Do you make a conclusion "This suggests that the quality of nadir L1 data is still poor" based on the fact that differences in tropospheric ozone are too large? This part needs to be revised.

*Author Response:*
*Agreed. We revised the text.*

5. Appendix. The Averaging Kernels for nadir observations change significantly with latitude and season. Showing just one example of the AK is not sufficient. I would suggest to show DFS profiles for different latitude bands and seasons, and specifically for the Antarctic latitudes in Sep-Oct. I also believe that this discussion belongs to section 2.2 where you should describe the main characteristics of the SCHIAMACHY nadir ozone retrievals.

*Author Response:*
*Following the suggestion of the reviewer, we added a new figure, Fig. 4, showing DFS profiles for different seasons for the years 2003 and 2009. See also response 6 to Referee #1.*

6. Section 5.1: On my opinion authors didn't provide enough evidences and explanations to demonstrate that version v8 is any better than v7_mfac.

*Author Response:*
*Indeed the differences between version 8 and version 7 with m-factors is not*

*large. However, there is a small improvement in using version 8, as can be seen from the standard deviations in Table 3 and the percentage differences in Table 4.*

7. Section 5.2: I believe that results presented in Fig. 7 and Table 5 are not enough to call this section 'Validation of v8'. For instance in section 7, authors speculate that on the days when the instrument was heated the measured radiances were affected. Have you tried to isolate and remove those days from your analysis? Do you see improvements/changes in the results?

*Author Response:*
*Our focus in the paper was to evaluate the different existing versions of L1 data. The referee makes a valid point of analysing and experimenting in detail with the radiometric and bias corrections. However this is beyond the scope of this paper. We reserve this to a future study. We limit ourselves in Sec. 5 to a discussion of potential future L1 improvements which should lead to better ozone profiles.*

8. Section 6: The analysis shown in this section is insufficient. There are many total ozone observations available for the considered time period that can be used to validate integrated ozone columns instead of looking at the reanalysis data. Also, the statement in the conclusion "we investigated the Antarctic ozone profile behavior in the austral spring season" doesn't correspond to the work shown in Section 6. There are many satellite ozone observations that overlap (or partially overlap) with SCHIAMACHY mission like SBUV/2 NOAA-17, Odin OSIRIS, ACE-FTS, Aura MLS and MIPAS. Comparisons with these correlative measurements would help you to understand how well SCHIAMACHY nadir profiles can describe the vertical ozone distribution in- side the ozone hole, and therefore if this dataset is suitable for studying inter-annual ozone variability over Antarctica. Without this extensive analysis it is not possible to claim that SCHIAMACHY nadir ozone profiles can be applied for the scientific analysis.

[Figure]

[Figure]

*Author Response:*
*Since we omitted the Antarctic ozone analysis in the revised paper, this comment is not applicable anymore.*

**Minor correction/typos:**

1. Section 6 p 14 line 35 and page 15 lines 1-2: I don't see the cost function or # of iterations (that are not shown in Table 6!!!). Do you mean Table 7 here?

   *Author Response:*
   *We omitted the Antarctic ozone analysis, so this is not relevant anymore.*

2. page 5, line 29: Should be "This allows" instead of "This amounts"

   *Author Response:*
   *Corrected.*

3. Table 2, Pressure grid- it would be useful to see the pressure grid used in your retrieval algorithm

   *Author Response:*
   *The retrieval pressure grid is given in Table 2.*

4. Table 3. Please, in the Table caption make a connection to the corresponding Figure 4. Also, it's not clear what do you mean by # of pixels? Is it a number of profiles considered for this comparison? Why is it different for different versions of Level 1 data? Please, spell "n_inter" as "number of iterations" in the caption. Could you explain what does it mean "median n_iter for #pixels"? Do you mean median number of iterations for the considered pixels (Profiles)? It should be "Column 7: standard deviation of Column 6".

*Author Response:*
*We revised Table 3 and its caption according to the suggestions of the reviewer.*

5. Tables 4 and 5. Please, fix "Troposphere [1000-100] hPa" and "Stratosphere [100- 10] hPa". The values shown in columns 6 and 7: are these biases for integrated stratospheric/tropospheric columns or mean over specified altitude range?

*Author Response:*
*We corrected the "Troposphere [1000-100] hPa" and "Stratosphere [100-10] hPa". The values in columns 6 and 7 are medians over the specified altitude range.*

6. Table 6: Please, clarify what quantities are shown in columns 2-4. Do they show min and max numbers of O3 profiles used to calculate the daily zonal mean value? It says "Column 5: Median uncertainty in stratospheric ozone column". Is it a correct label? What did you use to determine the tropopause pressures? Also why did you show total ozone values on Figure 8 and not stratospheric columns?

*Author Response:*
*Since we omitted the Antarctic ozone analysis, this comment is not applicable anymore.*

7. Table 7. Please, explain what is "relative uncertainties in ozone layer per height". Section 6, Figure 8 and Table 6 show results for total ozone columns. The numbers for sigma shown in Tables 6 and 7 are not the same. Please, add clear explanation of results shown in these tables.

*Author Response:*
*See response above.*

---

## Author Comment (AC4) · 25 Nov 2017

In the zip file attached to the supplement there are 5 files in total. They are: 1-Response to referee 1 2- Response to referee 2 3- Response to referee 3 4- Revised clean manuscript 5-Coloured pdf difference between old version and the new version

Please also note the supplement to this comment: https://www.atmos-meas-tech-discuss.net/amt-2017-136/amt-2017-136-AC4-supplement.zip

---

## Referee Report (RR1)

| Principal criteria | Excellent (1) | Good (2) | Fair (3) | Poor (4) |
|---|---|---|---|---|
| **Scientific significance:** Does the manuscript represent a substantial contribution to scientific progress within the scope of Atmospheric Measurement Techniques (substantial new concepts, ideas, methods, or data)? | | x | | |
| **Scientific quality:** Are the scientific approach and applied methods valid? Are the results discussed in an appropriate and balanced way (consideration of related work, including appropriate references)? | | x | | |
| **Presentation quality:** Are the scientific results and conclusions presented in a clear, concise, and well-structured way (number and quality of figures/tables, appropriate use of English language)? | | x | | |

I. General impression

In comparison with the first submitted version of this manuscript, this second version has much improved in many ways. The title, abstract, research intentions and presentation are clear and the work is well motivated, the scientific language and English formulations have strongly improved, and the unclarities in the original version have mostly been addressed and removed or elucidated. Yet two aspects of the paper might still require minor revision (see specific comments below).

II. Specific comments:

1.  This work addresses many aspects of SCIAMACHY L1 data and their evaluation in several ways. This certainly has scientific value, but too much information reduces the overall clarity. I would therefore suggest to leave out section 3 and the right column of Figure 3, and integrate the information on the left column of Figure 3 into section 4.1 (and combine Tables 3 and 4 correspondingly). A separate focus on the 10°N to 10°S L1 version effects on L2 seems somewhat redundant when all versions are compared with ozonesonde measurements on a global scale in a later section.

2.  It is unexplained and very confusing that the prior comparison results in terms of median bias are typically much better in the UTLS and below for the prior profiles than for the retrieved profiles! This might be due to the fact that the McPeters-Labow climatology is generated from ozonesonde (and MLS) profiles. In terms of random uncertainty however (i.e. comparison spread) the satellite retrievals might do better than the prior profiles due to their better caption of specific ozone dynamics. It is therefore suggested to either leave out the prior comparison (easy option) or add the comparison spread to the plots and explain the differences between satellite retrieval and prior comparisons bias and spread (scientifically preferred option).

III. Technical corrections:

In the abstract it is not fully clear whether the improvement suggestions made in the last paragraph are there for information only, or have really been tested. Please specify that the latter is true.

Page 1 line 61: When referring to Staehelin, please mention 70 "operational" stations worldwide. There are more stations, but not all provide data all the time.

Page 2 line 100: Remove double brackets around reference.

Section 3 and Figure 4: DFS profiles have little quantitative information if the corresponding layer thicknesses are unknown. Please explicitly refer to level-DFS or layer-DFS in the text and use markers instead of lines in the plot, or convert to DFS/km values for clarity.

Page 12 last paragraph of section 4.2 and Figure 8: "The validation results are clearly less noisy and smoother for the case where the AK was applied to the ozone sondes." is still rather vague. Please mention explicitly that AK-smoothing of the reference data is performed with the intention to remove the vertical smoothing difference error from the comparison error budget between satellite and ground-based observations. With this explicit formulation, Figure 8 even can be considered superfluous and can be left out of the manuscript, as it contains little information that contributes to the remainder of the discussion.